# FEATURE SELECTION USING STOCHASTIC GATES

## ABSTRACT

Feature selection problems have been extensively studied in the setting of linear estimation, for instance LASSO, but less emphasis has been placed on feature selection for non-linear functions. In this study, we propose a method for feature selection in non-linear function estimation problems. The new procedure is based on directly penalizing the $\ell_0$ norm of features, or the count of the number of selected features. Our $\ell_0$ based regularization relies on a continuous relaxation of the Bernoulli distribution, which allows our model to learn the parameters of the approximate Bernoulli distributions via gradient descent. The proposed framework simultaneously learns a non-linear regression or classification function while selecting a small subset of features. We provide an information-theoretic justification for incorporating Bernoulli distribution into our approach. Furthermore, we evaluate our method using synthetic and real-life data and demonstrate that our approach outperforms other embedded methods in terms of predictive performance and feature selection.

## 1 INTRODUCTION

Feature selection is a fundamental task in machine learning and statistics. Feature selection leads to a number of potential benefits: reducing experimental costs Min et al. (2014), enhancing interpretability (Ribeiro et al., 2016), computational speed up and even improving model generalization on unseen data (Chandrashekar & Sahin, 2014). In biomedicine, scientists collect multitude datasets comprising of many biomarkers (e.g., genes or proteins) that require development of effective diagnostics or prognostics models. For instance, in Genome wide association studies (GWAS), feature selection can help identify such models and lead to improved risk assessment and reduced cost.

Feature selection methods may be classified into three major categories: filter methods, wrapper methods, and embedded methods. Filter methods attempt to remove irrelevant features prior to learning a model. These methods filter features using a per-feature relevance score that is created based on some statistical measure (Battiti, 1994; Peng et al., 2005; Estévez et al., 2009; Song et al., 2007; 2012; Chen et al., 2017). Wrapper methods (Kohavi & John, 1997b; Stein et al., 2005; Zhu et al., 2007; Reunanen, 2003; Allen, 2013) use the outcome of a classifier to determine the relevance of each feature, which requires recomputing the classifier for each subset of features. This becomes computationally expensive for neural network based wrapper methods (Verikas & Bacauskiene, 2002; Kabir et al., 2010; Roy et al., 2015).

Embedded methods aim to remove this burden by learning the model while simultaneously selecting the subset of relevant features. The Least Absolute Shrinkage and Selection Operator (LASSO) (Tibshirani, 1996) is a well-known embedded method, whose objective is to minimize the loss while enforcing an $\ell_1$ constraint on the weights of the features. Although LASSO is scalable and widely used (Hans, 2009; Li et al., 2011; 2006), it is restricted to the domain of linear functions. To allow the model to capture nonlinear interaction, it is appealing to consider the non-convex extension of the LASSO formulation using neural networks.

We develop a fully *embedded feature selection* method for nonlinear models. To the best of our knowledge it is the first $\ell_0$ *embedded feature selection* method. Our method improves upon the LASSO formulation in two aspects: a) it captures nonlinear interactions between the features via neural network modeling and b) it employs an $\ell_0$-like regularization using gates whose weights are parametrized by a smooth variant of a Bernoulli distribution. Altogether these twofold improvements are formulated as a fully differentiable neural network. Specifically, our contributions are as follows:

1. By utilizing a recent development of continuous and differentiable approximation to discrete distributions (Maddison et al., 2016), (Jang et al., 2017), (Louizos et al., 2017), we introduce a solution to the long standing problem of feature selection with an $\ell_0$ regularization.

2. We present a novel simple relaxation of Bernoulli distribution to sparsify the input layer (the feature space) which we call stochastic gate (STG) and show its advantage over the distribution presented by (Louizos et al., 2017) both in performance and convergence time.

3. By applying these two relaxations to an input layer of a neural network, we perform embedded feature selection in classification, regression or survival analysis tasks and demonstrate its capabilities on artificial and real data sets.

4. We justify our probabilistic approach by analyzing the constrained Mutual Information maximization objective of feature selection. We demonstrate the applicability of our method using numerous examples (see Section 6 and Appendix).

**Notation:** We refer to vectors as bold lowercase $\boldsymbol{x}$ and random vectors as bold uppercase letters $\boldsymbol{X}$. Scalars are non-bold case $y$, while random variables are capital case $Y$. A set is represented by script fonts $\mathcal{X}, \mathcal{Y}, \mathcal{S}$. For example the $n^{th}$ vector-valued observation is denoted as $\boldsymbol{x}_n$ whereas $X_d$ represents the $d^{th}$ feature of the vector-valued random variable $\boldsymbol{X}$. Let $[n] = 1, 2, \ldots, n$. For a set $\mathcal{S} \subset [D]$ let the vector $\boldsymbol{s} \in \{0,1\}^D$ be the characteristic function for the set. That is $s_i = 1$ if $i \in \mathcal{S}$ and 0 otherwise. For two vectors $\boldsymbol{x}$ and $\boldsymbol{z}$ we denote $\boldsymbol{x} \odot \boldsymbol{z}$ to be the element-wise product between $\boldsymbol{x}$ and $\boldsymbol{z}$. Thus, if we let $\boldsymbol{s} \in \{0,1\}^D$ be the characteristic vector of $\mathcal{S}$, then we may define $\boldsymbol{x}_{\mathcal{S}} = \boldsymbol{x} \odot \boldsymbol{s}$. The $\ell_1$ norm of a vector is denoted as $\|\boldsymbol{x}\|_1 = \sum_{i=1}^{D} |x_i|$. Finally, the $\ell_0$ norm of a vector is denoted as $\|\boldsymbol{x}\|_0$ and counts the total number of non-zero entries in the vector $\boldsymbol{x}$.

## 2 PROBLEM SETUP AND BACKGROUND

Let $\mathcal{X} \in \mathbb{R}^D$ be the input domain with corresponding response domain $\mathcal{Y}$. Given realizations from some unknown data distribution $P_{X,Y}$, the goal of embedded feature selection methods is to simultaneously find a subset of indices $\mathcal{S} \subset \{1, ...D\}$ and construct a model that predicts $Y$ based on the selected features $\boldsymbol{X}_{\mathcal{S}}$.

### 2.1 RISK MINIMIZATION OBJECTIVE

We assume that we are given a family of functions $\mathcal{F}$ such that any function $f_{\boldsymbol{\theta}} \in \mathcal{F}$ is indexed by a set of parameters $\boldsymbol{\theta}$. Given some loss $L$, a selection of features $\mathcal{S} \subset [D]$, and a choice of parameters $\boldsymbol{\theta}$, we denote the risk of our model as

$$R(\boldsymbol{\theta}, \boldsymbol{s}) = \mathbb{E}_{X,Y} L(f_{\theta}(\boldsymbol{X} \odot \boldsymbol{s}), Y), \tag{1}$$

where we recall that $\boldsymbol{s} = \{0,1\}^D$ is a vector of indicator variables for the set $\mathcal{S}$ and $\odot$ denotes the point-wise product. Thus, the goal of the feature selection problem is to find the parameters $\boldsymbol{\theta}$ and $\boldsymbol{s}$ that minimize $R(\boldsymbol{\theta}, \boldsymbol{s})$ such that $\|\boldsymbol{s}\|_0$ is small compared to $D$.

### 2.2 FEATURE SELECTION FOR LINEAR MODELS

Before proceeding with our proposed method, we review the feature selection problem in the linear regression setting with the least squares loss. Thus, we restrict $\mathcal{F}$ to be the space of linear functions and the loss function to be the quadratic loss.

Given observations $\{\boldsymbol{x}_n, y_n\}_{n=1}^N$ we may consider the constrained empirical risk minimization problem

$$\min_{\boldsymbol{\theta}} \frac{1}{N} \sum_{n=1}^{N} L(\boldsymbol{\theta}^T \boldsymbol{x}_n, y_n) \quad \text{s.t. } \|\boldsymbol{\theta}\|_0 \leq k. \tag{2}$$

Since the above problem is intractable, a number of authors replace the $\ell_0$ constraint with a surrogate function $\Omega(\boldsymbol{\theta}) : \mathbb{R}^D \to \mathbb{R}_+$ designed to penalize the number of selected features in $\boldsymbol{\theta}$. A natural choice for $\Omega$ is the $\ell_1$ norm, which yields a convex problem and more precisely the LASSO optimization problem. The $\ell_1$ is known to be the closest convex relaxation to the $\ell_0$. In fact, in certain settings the

LASSO optimization and the $\ell_0$ based objective have the same solution. While the original LASSO problem focuses on the constrained optimization problem, the regularized least squares problem, which is often used in practice, yields the following minimization objective:

$$\min_{\boldsymbol{\theta}} \frac{1}{N} \sum_{n=1}^{N} (\boldsymbol{\theta}^T \boldsymbol{x}_n - y_n)^2 + \lambda \|\boldsymbol{\theta}\|_1. \tag{3}$$

The hyperparameter $\lambda$ trades off the amount of regularization versus the fit of the objective[1]. The $\ell_1$ regularized method is very effective for feature selection and prediction; however, it achieves this through shrinkage of the coefficients. As a result, (Fan & Li, 2001) have considered non-convex choices for $\Omega$ that perform well both theoretically and empirically for prediction and feature selection.

Our goal is to apply such regularization techniques to perform feature selection while learning a non-linear function. Kernel methods have been considered (Yamada et al., 2014), but scale quadratically in the number of observations. To alleviate this burden, (Gregorová et al., 2018) use random Fourier features to approximate the kernel. (Li et al., 2016) and (Scardapane et al., 2017) take an alternative approach by modeling $f_{\boldsymbol{\theta}}$ using a neural network with $\ell_1$ regularization on the input weights. However, in practice, introducing an $\ell_1$ penalty into gradient descent does not provide sufficient sparsification. Below, we discuss our method that works to directly use an $\ell_0$ penalty.

## 3 PROPOSED METHOD

We take a probabilistic approach to approximate the $\ell_0$ norm, which can extend to non-linear models while remaining computationally efficient. To motivate such probabilistic approach, we provide theoretical support (see Section 4) based on a Mutual Information perspective of the feature selection problem.

To view the $\ell_0$ regularized version of the risk (Eq. 1) from a probabilistic perspective, one can introduce a Bernoulli random vector $\tilde{\boldsymbol{S}}$ whose entries are independent and the $d^{th}$ entry satisfies $\pi_d = \mathbb{P}(\tilde{S}_d = 1)$ for $d \in [D]$. If we denote the empirical expectation over our observations as $\hat{\mathbb{E}}_{X,Y}$, then, the empirical regularized risk (Eq. 1) becomes

$$\min_{\boldsymbol{\theta}, \boldsymbol{\pi}} \hat{R}(\boldsymbol{\theta}, \boldsymbol{\pi}) = \min_{\boldsymbol{\theta}, \boldsymbol{\pi}} \hat{\mathbb{E}}_{X,Y} \mathbb{E}_{\tilde{S}} \left[ L(f_\theta(\boldsymbol{X} \odot \tilde{\boldsymbol{S}}), Y) + \lambda \|\tilde{\boldsymbol{S}}\|_0 \right], \tag{4}$$

where we have $\mathbb{E}_{\tilde{S}} \|\tilde{\boldsymbol{S}}\|_0 = \sum_{d=1}^{D} \pi_d$ and we constrain $\pi_d \in \{0, 1\}$. Clearly, this formulation is equivalent to Eq. 1, with a regularized penalty on cardinality rather than an explicit constraint. We may then relax the discrete constraint on $\pi_d$ to be $\pi_d \in [0, 1]$.

Now, our goal is to find the model parameters $\boldsymbol{\theta}^*$ and Bernoulli parameters $\boldsymbol{\pi}^*$ that minimize the empirical risk $\hat{R}(\boldsymbol{\theta}, \boldsymbol{\pi})$ via gradient descent. However, an optimization of a loss function which includes discrete random variables suffers from high variance (see F in the Appendix for more details). Therefore, inspired by a recently developed continuous approximation for discrete random variables, suggested by (Jang et al., 2017; Maddison et al., 2016), we develop and use a novel and simple continuous distribution that is fully differentiable and suited for the task of feature selection.

### 3.1 CONTINUOUS RELAXATION

Our continuous relaxation for the Bernoulli variables $\tilde{S}_d$ for $d \in [D]$ is termed stochastic gate (STG). The STG relies on the reparametrization trick, which is widely used for reducing the variance of gradient estimators (Miller et al., 2017; Figurnov et al., 2018). To construct a continuous approximation to Bernoulli random variable via the reparametrization trick, we define $z_d = g(\mu_d + \epsilon_d) = \max(0, \min(1, \epsilon_d + \mu_d + 0.5))$ where $\epsilon_d$ is drawn from a Gaussian distribution $\mathcal{N}(0, \sigma_d)$, where $\sigma_d$ is fixed throughout training. This approximation can be viewed as a clipped, mean-shifted, Gaussian random vector. Furthermore, the gradient of the objective with respect to $\mu_d$ can be computed via the chain rule.

We can now rewrite the objective in Eq. 4 as

$$\min_{\boldsymbol{\theta}, \boldsymbol{\mu}} \hat{R}(\boldsymbol{\theta}, \boldsymbol{\mu}) = \min_{\boldsymbol{\theta}, \boldsymbol{\mu}} \mathbb{E}_{X,Y} \mathbb{E}_Z \left[ L(f_\theta(\boldsymbol{X} \odot \boldsymbol{Z}), Y) + \lambda \|\boldsymbol{Z}\|_0 \right], \tag{5}$$

---

[1] $\lambda$ has a one-to-one correspondence to $k$ in the convex setting via Lagrangian duality.

where $\boldsymbol{Z}$ is a random vector with $D$ independent variables $z_d$ for $[D]$. To optimize the empirical surrogate of the objective (Eq. 5), we first differentiate it with respect to $\boldsymbol{\mu}$. Then, Monte Carlo sampling leads us to the following gradient estimator

$$\frac{\partial}{\partial \mu_d} \hat{R}(\boldsymbol{\theta}, \boldsymbol{\mu}) = \frac{1}{K} \sum_{k=1}^{K} \left[ L'(\boldsymbol{z}^k) \frac{\partial z_d^k}{\partial \mu_d} \right] + \frac{\lambda}{K} \frac{\partial}{\partial \mu_d} \sum_{k=1}^{K} \Pr\{z_d^k > 0\},$$

where $K$ is the number of Monte Carlo samples. Thus, we can update the parameters $\mu_d$ for $[D]$ via gradient descent. We note that if we replace $\frac{\partial z_d^k}{\partial \mu_d}$ with 1, the above gradient estimator for $L'$ is reduced to the Straight-Through estimator (Bengio et al., 2013).

Under the continuous relaxation, the expected regularization term in the objective $\hat{R}(\boldsymbol{\theta}, \boldsymbol{\mu})$ is simply the sum of the probability that the gates $\{z_d\}_{d=1}^{D}$ are active, which is equal to $\sum_{d=1}^{D} \Phi\left(\frac{\mu_d + \frac{1}{2}}{\sigma_d}\right)$, where $\Phi$ is the standard Gaussian CDF. To conclude, we can now optimize the objective in Eq. 5 using gradient descent over the model parameters $\boldsymbol{\theta}$ and the parameters $\boldsymbol{\mu}$ representing the Gaussian's mean (instead of the Bernoulli parameters $\boldsymbol{\pi}$).

After training, to remove the stochasticity from the learned gates, we set $\hat{z}_d = \max(0, \min(1, \mu_d + 0.5))$, which informs what features are selected. Note that when $|\mu_d|$ is less than $\frac{1}{2}$, $\hat{z}_d$ returns the value between $(0, 1)$. In such a case, we can treat the value of $\hat{z}_d$ as feature importance or employ an additional thresholding (i.e. 1 if $\hat{z}_d > 0.5$ and 0 otherwise) depending on application-specific needs. In the Appendix, we provide the pseudo-code of our algorithm as well as the discussion of the choice of $\sigma_d$.

## 4 CONNECTION TO MUTUAL INFORMATION

In this section we show an equivalence between the Bernoulli formulation of the feature selection problem and the $\ell_0$ regularized approach.

### 4.1 MUTUAL INFORMATION BASED OBJECTIVE

From an information theoretic perspective, the goal of feature selection is to find a subset of features $\mathcal{S}$ that has the highest Mutual Information (MI) with the target variable $Y$. Recall that the MI between two random variables can be defined as $I(\boldsymbol{X}; Y) = H(Y) - H(Y|\boldsymbol{X})$ where $H(Y), H(Y|\boldsymbol{X})$ are the entropy of $p_Y(Y)$ and the conditional entropy of $p_{Y|\mathbf{X}}(Y|\boldsymbol{X})$, respectively (Cover & Thomas, 2006). Then we can formulate the task as selecting $\mathcal{S}$ such that the mutual information between $\boldsymbol{X}_\mathcal{S}$ and $Y$ are maximized:

$$\max_{\mathcal{S}} I(\boldsymbol{X}_\mathcal{S}, Y) \quad \text{s.t. } |\mathcal{S}| = k, \tag{6}$$

where $k$ is the hypothesised number of relevant features.

### 4.2 INTRODUCING RANDOMNESS

We first demonstrate that under mild assumptions we can replace the deterministic search over the set $\mathcal{S}$ (or corresponding indicator vector $\boldsymbol{s}$), by a search over the parameters of the distributions that model $\boldsymbol{s}$. Our proposition is based on the following two assumptions:

**Assumption 1:** There exists a subset of indices $\mathcal{S}^*$ with cardinality equal to $k$ such that for any $i \in \mathcal{S}^*$ we have $I(X_i; Y|\boldsymbol{X}_{\backslash\{i\}}) > 0$.
**Assumption 2:** $I(\boldsymbol{X}_{\mathcal{S}^{*c}}; Y|\boldsymbol{X}_{\mathcal{S}^*}) = 0$.

**Discussion of assumptions:** Assumption 1 that including an element from $\mathcal{S}^*$ improves prediction accuracy. This assumption is equivalent to stating that feature $i$ is strongly relevant (Kohavi & John, 1997a; Brown et al., 2012). Assumption 2 simply states that $\mathcal{S}^*$ is a superset of the Markov Blanket of the variable $Y$ (Brown et al., 2012). The assumptions are quite benign. For instance they are satisfied if $\boldsymbol{X}$ is drawn from a Gaussian with a non-degenerate covariance matrix and $Y = f(\boldsymbol{X}_{\mathcal{S}^*}) + w$, where $w$ is noise independent of $\boldsymbol{X}$ and $f$ is not degenerate. With these assumptions in hand, we may present our result.

**Proposition 1.** *Suppose that the above assumptions hold for the model. Then, solving the optimization in Eq. 6 is equivalent to solving the optimization*

$$\max_{\mathbf{0} \leq \boldsymbol{\pi} \leq \mathbf{1}} I(\boldsymbol{X} \odot \tilde{\boldsymbol{S}}; \boldsymbol{Y}) \quad s.t. \quad \sum_i \mathbb{E}[\tilde{S}_i] \leq k, \tag{7}$$

*where the coordinates $\tilde{S}_i$ are drawn independently at random according to a Bernoulli distribution with parameter $\pi_i$.*

Due to length constraints, we leave the proof of this proposition and how it bridges the MI maximization (Eq. 6) and risk minimization (Eq. 2) to the Appendix (see Section C and D).

## 5 RELATED WORK

The two most related works to this study are (Louizos et al., 2017) and (Chen et al., 2018). In (Louizos et al., 2017), they introduce the Hard-Concrete (HC) distribution as a continuous surrogate for Bernoulli distributions in the context of model compression. The HC distribution is induced by applying a hard sigmoid $z = \min(1, \max(0, \bar{s}))$ to $\bar{s}$, where we construct $\bar{s}$ by applying Sigmoid function to the logistic distribution as follows:

$$u \sim U(0,1), L = \log(u) - \log(1-u), s = \frac{1}{1 + \exp(\frac{-(\log \alpha + L)}{\beta})}, \bar{s} = s(\zeta - \tau) + \tau$$

The interval $(\tau, \zeta)$, with $\tau < 0$ and $\zeta > 1$, allows the distribution to have more probability mass on the edge of the support. The authors demonstrate how incorporating the HC in deep neural networks leads to fast convergence and improved generalization due to the sparsification effect. In this study, we introduce this type of sparsification for the long standing problem of feature selection with $\ell_0$ regularization.We demonstrate that a simple relaxation of Bernoulli distributions is sufficient and works better than the HC distribution for feature selection. We find empirically that the HC yields high variance gradient estimates compared to the STG. This high-variance does not affect model sparsification as the sparsity pattern does not matter. Contrarily, for feature selection it is crucial as practitioners require a stable procedure. Furthermore, higher variance gradients will result in a slower convergence of the model (this has been demonstrated empirically in the supp. material). In (Chen et al., 2018), the Gumbel-softmax trick is used to develop a framework for interpreting pre-trained models. Their method is focused on finding a subset of features given a particular instance, and therefore is not appropriate for general feature selection.

Some studies tackle embedded feature selection problems by extending LASSO and group LASSO to neural network models. The authors (Li et al., 2016; Scardapane et al., 2017) and (Feng & Simon, 2017) have a similar goal as ours in performing feature selection, but instead rely on the $\ell_1$ relaxation to the $\ell_0$. Our approach, which utilizes stochastic gates coupled with the $\ell_0$ norm achieves substantially better empirical performance compared against other $\ell_1$ based baselines. This point is demonstrated in the next section.

## 6 EXPERIMENTS

Here we provide empirical evaluation of our method in a wide range of settings. We begin with simple linear regression evaluating the capabilities of our approach to select a sparse set of relevant features (section 6.1). We then evaluate our method in nonlinear regression tasks (section 6.2). Next, we demonstrate the applicability of the method in a highly nonlinear classification task (section 6.3. Finally, we present two potential applications (section 7).

We implement our method using both the STG and HC distributions and compare their applicability for feature selection. The hyperparamters of all the methods are optimized using validation sets via Optuna (Takuya Akiba & Koyama, 2019). In the Appendix, we provide all details of the Optuna based parameter tuning procedure. Additional experiments also appear in the Appendix, including high dimensional datasets and an extensive comparisons between the STG and HC (Louizos et al., 2017) distributions.

### 6.1 SUPPORT RECOVERY IN LINEAR REGRESSION

In the setting of noisy linear regression, (Wainwright, 2009) have analyzed the probability of LASSO to correctly identify a sparse subset of active variables. The problem is know as support recovery and is formulated as follows; let $\boldsymbol{\beta}^* \in \mathbb{R}^D$ be a fixed sparse vector, such that $\beta_i^* \in \{-0.5, 0.5\}$ (with equal probability) if $i \in \mathcal{S}$, and $\beta_i^* = 0$ otherwise. Suppose the cardinality of the support $|\mathcal{S}| = k$ is known. Given a matrix of measurements $\boldsymbol{X} \in \mathbb{R}^{N \times D}$ with values drawn independently from $N(0, 1)$, the response $\boldsymbol{y}$ is defined as

$$\boldsymbol{y} = \boldsymbol{X}\boldsymbol{\beta}^* + \boldsymbol{w}, \tag{8}$$

where the values of the noise $\boldsymbol{w}_i, i = 1, ..., N$ are drawn independently from $N(0, 0.5)$.

Here, we reproduce this setting to evaluate the probability of the proposed approach to perfectly recover the support of $\boldsymbol{\beta}^*$. As suggested by (Wainwright, 2009), we use a sparsity that scales with $D$ such that $k = \lceil 0.4D^{0.75} \rceil$. For each number of samples $N$ in the range $[10, 500]$, we run 200 simulations and count the portion of correctly recovered supports. We repeat this process for 2 different values of $D$ and compare our performance to LASSO. For LASSO, the regularization parameter was set to its optimal value $\alpha_N = \sqrt{\frac{2\sigma^2 \log(D-k) \log(k)}{N}}$ (Wainwright, 2009). For STG and HC we set $\lambda_N = C\alpha_N$, such that $C$ is a constant, which is selected using a grid search in the range [0.1,10]. As evident from Fig. 1, even when restricting to linear functions our method has a clear advantage over LASSO. This implies that the $\ell_0$ based penalty, although is non convex in nature, allows perfect recovery of the support using less samples. Furthermore, the proposed STG requires even less samples than the HC distribution and suffers from a smaller variance.

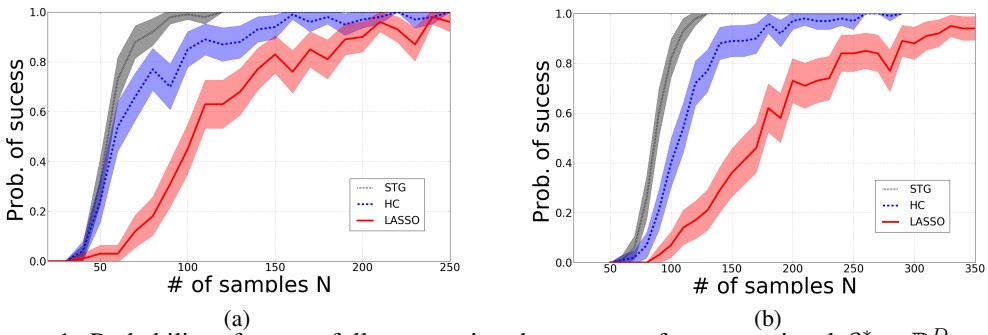

(a)                 (b)

Figure 1: Probability of successfully recovering the support of a sparse signal $\boldsymbol{\beta}^* \in \mathbb{R}^D$ vs. the number of observations $N$. Comparison between the proposed method, HC and LASSO under a linear additive noise model $\boldsymbol{y} = \boldsymbol{X}\boldsymbol{\beta}^* + \boldsymbol{w}$ using (a) $D = 64$ and (b) $D = 128$. Central lines are the means while shaded are represent the standard deviation.

### 6.2 NONLINEAR REGRESSION USING SYNTHETIC AND REAL DATASETS

In this section, we evaluate our method for regression tasks against two other embedded feature selection methods: LASSO and Sparse Random Fourier Features (Gregorová et al., 2018). Following the same format as (Gregorová et al., 2018), the following functions are used to generate synthetic data: (SE1: 100/5) $y = \sin(x_1 + x_3)^2 \sin(x_7 x_8 x_9) + \mathcal{N}(0, 0.1)$. (SE2: 18/5) $y = \log((\sum_{s=11}^{15} x_s)^2) + \mathcal{N}(0, 0.1)$. (SE3: 1000/10) $y = 10(z_1^2 + z_3^2)e^{-2(z_1^2 + z_3^2)} + \mathcal{N}(0, 0.01)$, where each $x_i$ is drawn from the standard Gaussian $\mathcal{N}(0, 1)$. For (SE3), the five consecutive coordinates are generated by $x_{5(j-1)+i} = z_j + \mathcal{N}(0, 0.1)$, where $z_j \sim \mathcal{N}(0, 1)$ for $i = 1, ..., 5$ and $j = 1, ..., 200$. The numbers next to the experiment code indicate total dimensions/relevant dimensions in the feature space. We also evaluate our method using three real datasets: (RCP: 21/-), of computer systems activity, (REL 17/-) of F16 elevators and (RAI 39/-) of F16 ailernos all taken from the LIACC repository [2]. For each dataset, we generate 30 different replications and randomly split the data into train, validation, and test set (see Appendix for more details). The root mean squared error on the test set averaged over 30 random replicated datasets are reported in Table 1. Our method outperforms all alternative methods for most cases. Note that (SE1) is generated using a sine function, which is in favor of the random Fourier feature based method (SRFF).

---

[2]http://www.dcc.fc.up.pt/~ltorgo/Regression/DataSets.html

Table 1: Regression performance comparison in terms of root mean squared error. The mean and standard deviation are computed across 30 resamples. $N$ is 1000 for SE1,2,3 and 6000 for RCP. The values for LASSO and SRFF are borrowed from (Gregorová et al., 2018). DNN represents a deep neural network without feature selection.

| EXP | LASSO | RF | SG-L1-NN | SRFF | DNN | HC | STG |
|-----|-------|-----|----------|------|-----|-----|-----|
| SE1 | 0.29 (0.01) | 0.30 (0.01) | 0.29 (0.01) | **0.27** (0.01) | 0.29 (0.01) | 0.29 (0.01) | 0.29 (0.01) |
| SE2 | 2.22 (0.10) | 2.34 (0.17) | 2.35 (0.18) | 1.60 (0.10) | 2.05 (0.11) | 1.19 (0.31) | **0.87** (0.15) |
| SE3 | 0.68 (0.002) | 0.50(0.01) | 0.68 (0.01) | 0.48 (0.03) | 0.73 (0.02) | 0.33 (0.05) | **0.14** (0.10) |
| RCP | 9.69 (0.71) | 3.52 (0.12) | 9.64 (0.65) | 2.52 (0.18) | 2.89 (0.25) | 2.74 (0.75) | **2.44** (0.08) |
| REL | 0.47 (0.01) | 0.58 (0.01) | 0.44 (0.01) | 0.31 (0.03) | 0.61 (0.01) | 0.33 (0.04) | **0.27** (0.002) |
| RAI | 0.43 (0.02) | 0.48 (0.003) | 0.47(0.002) | 0.41(0.02) | 0.44 (0.01) | 0.41 (0.01) | **0.39** (0.01) |

## 6.3 Noisy Binary XOR Classification

In the following evaluation, we consider the problem of learning a binary XOR function for classification task. The first two coordinates $x_1, x_2$ are drawn from a binary "fair" Bernoulli distribution. The response variable is set as an XOR of the first coordinates, such that $y = x_1 \oplus x_2$. The coordinates $x_i, i = 3, ..., D$ are nuisance features, also drawn from a binary "fair" Bernoulli distribution. The number of points we generate is $N = 1,500$, of which 70 % are reserved for test and 10% of the remaining training set was reserved for validation. We compare the proposed method to four embedded feature selection methods (LASSO (Tibshirani, 1996), C-support vectors (SVC) (Chang & Lin, 2011), deep feature selection (DFS) (Li et al., 2016), sparse group regularized NN (SG-L1-NN) (Scardapane et al., 2017)). To provide more benchmarks, we also compare our embedded method against three wrapper methods (Extremely Randomized Trees (Tree) (Rastogi & Shim, 2000), Random Forests (RF) (Strobl et al., 2008)) and Extreme Gradient Boosting (XGBOOST) (Chen & Guestrin, 2016).

To evaluate the feature selection performance, we calculate the Informative Features Weight Ratio (IFWR). IFWR is defined as the sum of weights $W_d$ over the informative features divided by the sum over all weights. In the case of binary weights the IFWR is in fact a recall measure for the relevant features (See the Appendix for more details.)

The experiment is repeated 20 times for different values of $D$, and the average test classification accuracy and standard deviation are presented in Fig. 2(a), followed by the IFWR in Fig. 2(b). The number of selected features affects the accuracy. Therefore, to treat all the methods in a fair manner, we tune the hyperparameter that controls the sparsity level using Optuna (Takuya Akiba & Koyama, 2019) which optimizes the overall accuracy across different $D$s. For instance, the wrapper methods (Tree, RF and XGBOOST) has a threshold value to retain features. We retrain them using only such features whose weight is higher than the threshold. In terms of feature ranking (see Fig. 2(c), only the tree based methods and the proposed (STG and HC based) provide the optimal median rank (which is 1.5) for the two relevant features. Nonetheless, the ranking provided by STG is the most stable comparing to all the alternative methods.

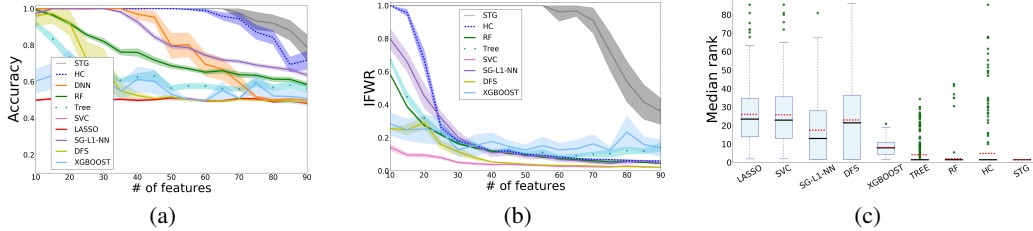

(a)                                (b)                                (c)

Figure 2: (a) Classification accuracy (mean and standard deviation) vs. the number of irrelevant noisy dimension ($D$) for the XOR problem. (b) The Informative Features Weight Ratio (IFWR), central lines are the means while shaded are represent the standard deviation. IFWR is the sum of weights $W_d$ over the informative features divided by the sum over all weights. (c) Box plots for the median rank of the two informative features. Black line and dashed red represent the median and mean of each method. Optimal median rank in this experiment is 1.5.

## 7 Application

### 7.1 Purified populations of peripheral blood monocytes (PBMCs)

In (Zheng et al., 2017), the authors have subjected more than 90,000 purified cell populations of peripheral blood monocytes (PBMCs) to Single-cell RNA sequencing (scRNA-seq) analysis. Here

Table 2: Performance comparison of survival analysis on METABRIC. We run the same experiment 5 times with different train/test split and report the mean and the standard deviation on the test set. In (Katzman et al., 2018), it is reported that DeepSurv outperforms other existing survival analysis methods such as Random Survival Forest (RSF) (Ishwaran et al., 2008) and the original Cox Propotional Hazard Model.

|  | DEEPSURV | RSF | COX-LASSO | COX-HC | COX-STG |
|---|---|---|---|---|---|
| C-INDEX | 0.612 (0.009) | 0.626 (0.006) | 0.580 (0.003) | **0.636** (0.007) | 0.633 (0.005) |
| # FEATURES | 221 (ALL) | 221 (ALL) | 44 (0) | 8 (0.89) | 2 (0) |

we use this data and focus on classifying two subpopulations of T-cells, namely Naive and regulatory T-cells. We use the proposed method to select a subset of genes for which the network discriminates between Naive and regulatory T-cells. We first filter out the genes that are lowly expressed in the cells, which leaves us with $D = 2538$ genes (features). The total number of cells in these two classes is $N = 20742$, of which we only use $10\%$ of the data for training. We apply the proposed method for different values of $\lambda$ and report the number of selected features and classification accuracy on the test set. Here we compare our performance (STG and HC) to RF and LASSO. A least squares polynomial fit plot of the accuracy vs. number of selected features is presented in Fig. 3. The accuracy obtained by a NN without feature selection is $91.09\%$, which is comparable to what we achieve with a small fraction of the features. We have also evaluated the performance of the Hard-Concrete applied to all layers (HC-Full), following the procedure in (Louizos et al., 2017). Empirically we observed that using this type of regularization across all layers provides inferior capabilities in terms of feature selection. Moreover, when the HC-Full converges to a larger subset of features ($d > 50$), it does not generalize at all and the test accuracy is around $0.5$.

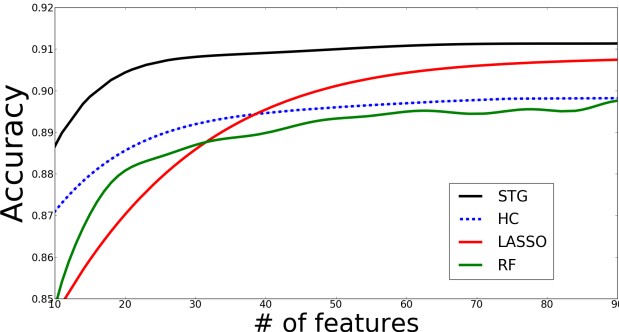

Figure 3: Classification of T-cells sub-populations. Accuracy vs. number of features selected by each method. Comparison of the proposed method (STG and HC) to Random Forests (RF) and LASSO.

## 7.2 COX PROPORTIONAL HAZARD MODELS FOR SURVIVAL ANALYSIS

We incorporate our STG into DeepSurv (Katzman et al., 2018) to examine how this procedure improves survival analysis. The input to this analysis are gene expression profiles (Curtis et al., 2012) along with additional commonly used clinical variables, the goal is to predict survival time. In this setting, identifying a small subset of predictive covariates is important as it may lead to cheaper and more stable medical assays. See the Appendix for additional description of the data and the experimental setup.

We compared our method (Cox-STG and COX-HC) against three other methods: Cox model with $\ell_1$ regularization (Cox-LASSO), Random Survival Forest (RSF) (Ishwaran et al., 2008) and the original DeepSurv. We evaluate the predictive ability of the learned models based on the concordance index (CI), a standard performance metric for model assessment in survival analysis, which measures the agreement between the rankings of the predicted and observed survival times. The performance in terms of the CI is reported in Table 2. We see that Cox-HC and Cox-STG outperform the alternative methods, indicating that our method shrinks the feature size while learning a predictive model.

## 8 CONCLUSION

In this paper, we propose a novel embedded feature selection method for neural networks and linear models based on stochastic gates. It has an advantage over previous $\ell_1$ based regularization methods in terms of achieving a high level of sparsity while learning effective non linear models. We justify our probabilistic feature selection framework from the information theoretic perspective. In experiments, we demonstrate that our method consistently outperforms existing embedded feature selection methods in both synthetic and real datasets.

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

## A    APPENDIX

## B    ALGORITHM

Here we present the pseudo code of our method, presented in Algorithm 1. The loss $L$ is typically negative-log likelihood for classification and squared loss for regression. $N$ is the sample size, $D$ is the number of features, $K$ is the number of Monte Carlo samples.

**Input:** $\boldsymbol{X} \in \mathbb{R}^{N \times D}$, target variables $\boldsymbol{y} \in \mathbb{R}^N$, regularization parameter $\lambda$, number of epochs $M$, learning rate $\gamma$,
**Output:** Trained model $f_{\boldsymbol{\theta}}$ and parameter $\boldsymbol{\mu} \in \mathbb{R}^D$.
 1: Initialize the model parameter $\boldsymbol{\theta}$. Set $\boldsymbol{\mu} = 0$.
 2: **for** $i = 1, ..., M$ **do**
 3:     **for** $n = 1, ..., N$ **do**
 4:         **for** $d = 1, ..., D$ **do**
 5:             **for** $k = 1, ..., K$ **do**
 6:                 Sample $\epsilon_d^{(k)} \sim N(0, \sigma_d)$
 7:                 Compute the gate $z_d^{(k)} = \max(0, \min(1, \mu_d + \epsilon_d^{(k)} + 0.5))$
 8:             **end for**
 9:             Set $z_d = \frac{1}{K} \sum_{k=1}^{K} z_d^{(k)}$
10:         **end for**
11:         Set $\boldsymbol{z} = [z_1, ..., z_D]^T$
12:     **end for**
13:     Compute the loss $\hat{L} = \frac{1}{N} \sum_{n=1}^{N} L(f_\theta(\boldsymbol{x}_n \odot \boldsymbol{z}), y_n)$
14:     Compute the regularization term $R = \lambda \sum_{d=1}^{D} \Phi(\frac{\mu_d + 0.5}{\sigma_d})$
15:     Update $\boldsymbol{\theta} := \boldsymbol{\theta} - \gamma \nabla_{\boldsymbol{\theta}} \hat{L}$ and $\boldsymbol{\mu} := \boldsymbol{\mu} - \gamma \nabla_{\boldsymbol{\mu}} (\hat{L} + R)$
16: **end for**
17:

**Algorithm 1:** STG: Feature selection using stochastic gates

We empirically observe that setting the number of Monte Carlo samples $K = 1$ and the standard deviation of the Gaussian distribution $\sigma_d = 0.5$ for $d = 1, ..., D$ suffices for feature selection in our experiments. See Section E in the Appendix for more details about the specific choice of $\sigma_d$. After training, the set of indices for selected features is: $\{d : \min(1, \max(0, \mu_d + 0.5)) > 0\}$.

## C    PROOF OF PROPOSITION 1

We now provide a proof for Proposition 1, showing the equivalence between the stochastic optimization (Eq.7) and the deterministic one (Eq. 6). Let $\tilde{\mathcal{S}}$ be a subset such that $\mathcal{S}^* \setminus \tilde{\mathcal{S}} \neq \emptyset$. That is there exists some element in $\mathcal{S}^*$ that is not in $\tilde{\mathcal{S}}$. For any such set $\tilde{\mathcal{S}}$ we have that $I(\boldsymbol{X}_{\tilde{\mathcal{S}}}; Y) < I(\boldsymbol{X}; Y)$. Indeed, if we let $i \in \mathcal{S}^* \cap \tilde{\mathcal{S}}^c$ then we have

$$
\begin{aligned}
I(\boldsymbol{X}_{\tilde{\mathcal{S}}}; Y) &\leq I(\boldsymbol{X}_{\setminus \{i\}}; Y) \\
&= I(\boldsymbol{X}; Y) - I(\boldsymbol{X}_i; Y | \boldsymbol{X}_{\setminus \{i\}}) \\
&< I(\boldsymbol{X}; Y),
\end{aligned}
$$

where the final inequality follows by Assumption 1. Assumption 2 also yields that for any set $\tilde{\mathcal{S}}$ such that $S^* \subset \tilde{\mathcal{S}}$, we have $I(\boldsymbol{X}_{\tilde{\mathcal{S}}}; Y) = I(\boldsymbol{X}; Y)$. Now, when we consider the Bernoulli optimization problem we have

$$
\max_{\boldsymbol{\pi}} I(\boldsymbol{X} \odot \boldsymbol{S}; \boldsymbol{Y}) \quad \text{s.t.} \quad \sum_l \boldsymbol{\pi}_l \leq k \text{ and } 0 \leq \boldsymbol{\pi}_l \leq 1.
$$

The mutual information can be expanded as

$$
I(\boldsymbol{X} \odot \boldsymbol{S}; Y) = \sum_s I(\boldsymbol{X} \odot \boldsymbol{s}; Y) p_{\boldsymbol{\pi}}(\boldsymbol{S} = \boldsymbol{s}),
$$

where we have used the fact that $\boldsymbol{S}$ is independent of everything else. Recall that in optimization (Eq. 7) the coordinates of $\boldsymbol{S}$ are sampled at random. Therefore, the distribution that is being optimized over $p_{\boldsymbol{\pi}}$ is a product distribution. Our goal is to understand the form of this distribution. To that end, we will consider a problem dropping the independence constraint. If we can show that the distribution found by solving this new optimization problem with less constraints is still a product distribution, then we obtain a solution to the original optimization (Eq. 6).

Now, from above we know that the optimal value of the optimization is $I(\boldsymbol{X} \odot \tilde{\boldsymbol{S}}; Y)$ for any set $\mathcal{S}^* \subset \tilde{\mathcal{S}}$. Hence, any unconstrained distribution should place all of its mass on such subsets in order to maximize the mutual information. As a result $\sum_{l \in S^*} p(\boldsymbol{S}_l = 1) = k$. However, there is an optimization constraint that $\mathbb{E}[\sum_l \boldsymbol{S}_l] \leq k$. Therefore, $\mathbb{E}[\boldsymbol{S}_l] = 0$ for any $l \notin \mathcal{S}^*$. Hence, the optimal solution is to select the distribution so that all of the mass is placed on the subset $S^*$ and no mass elsewhere. As this is also a product distribution, this complete the proof of the claim.

## D    BRIDGING THE TWO PERSPECTIVES

To motivate the introduction of randomness into the risk, we have looked at the feature selection problem from a MI perspective. Based on the MI objective, we have observed that introducing randomness into the constrained maximization procedure, does not change the objective (Proposition 1). Here we provide a relation between the MI objective (Eq. 6) to the empirical risk (Eq. 1), which supports our proposed procedure.

We first note that the MI maximization over the set $\mathcal{S}$ can be reformulated as the minimization of the conditional entropy $H(Y|\boldsymbol{X}_{\mathcal{S}})$ since $H(Y)$ does not depend on $\mathcal{S}$:

$$\max_{\mathcal{S}} I(\boldsymbol{X}_{\mathcal{S}}; Y) = \max_{\mathcal{S}} H(Y) - H(Y|\boldsymbol{X}_{\mathcal{S}}) \iff \min_{\mathcal{S}} H(Y|\boldsymbol{X}_{\mathcal{S}}).$$

Recall that $\boldsymbol{X}_{\mathcal{S}} = \boldsymbol{X} \odot \tilde{\boldsymbol{S}}$. By Proposition 1, we can rewrite the deterministic search over the set $\mathcal{S}$ by a search over the Bernoulli parameters $\boldsymbol{\pi}$:

$$\min_{\boldsymbol{\pi}} H(\boldsymbol{Y}|\boldsymbol{X} \odot \tilde{S}) = \min_{\boldsymbol{\pi}} \mathbb{E}_{X,Y,\tilde{S}} - \log P_{\boldsymbol{\theta}^*}(Y|\boldsymbol{X} \odot \tilde{S})$$
$$= \min_{\boldsymbol{\theta}} \min_{\boldsymbol{\pi}} \mathbb{E}_{X,Y,\tilde{S}} - \log P_{\boldsymbol{\theta}}(Y|\boldsymbol{X} \odot \tilde{S}),$$

where the expectation is over $\boldsymbol{X}, Y \sim P_{\boldsymbol{\theta}_*}$, which is the true data distribution, and $\tilde{S} \sim Bern(\tilde{S}|\boldsymbol{\pi})$. Put our model distribution as $P_\theta$, then we can rewrite the right hand side as:

$$\mathbb{E}_{X,Y,\tilde{S}} \log P_{\theta^*}(Y|\boldsymbol{X} \odot \tilde{\boldsymbol{S}}) = \mathbb{E}_{X,Y,\tilde{S}} \left[ \log P_{\theta^*}(Y|\boldsymbol{X} \odot \tilde{\boldsymbol{S}}) \frac{P_\theta(Y|\boldsymbol{X} \odot \tilde{\boldsymbol{S}})}{P_\theta(Y|\boldsymbol{X} \odot \tilde{\boldsymbol{S}})} \right]$$
$$= \mathbb{E}_{X,Y,\tilde{S}} \log \frac{P_{\theta*}(Y|\boldsymbol{X} \odot \tilde{\boldsymbol{S}})}{P_\theta(Y|\boldsymbol{X} \odot \tilde{\boldsymbol{S}})} + \mathbb{E}_{X,Y,\tilde{S}} \log P_\theta(Y|\boldsymbol{X} \odot \tilde{\boldsymbol{S}}).$$

Since $\text{KL}(P_{\theta*}(Y|\boldsymbol{X} \odot \tilde{\boldsymbol{S}})||P_\theta(Y|\boldsymbol{X} \odot \tilde{\boldsymbol{S}}))$ is non-negative, $\mathbb{E}_{\tilde{S}}\text{KL}(P_{\theta*}(Y|\boldsymbol{X} \odot \tilde{\boldsymbol{S}})||P_\theta(Y|\boldsymbol{X} \odot \tilde{\boldsymbol{S}}))$ is also non-negative because it is a weighted sum of non-negative terms. Noting that

$$\mathbb{E}_{\tilde{S}}\text{KL}(P_{\theta*}(Y|\boldsymbol{X} \odot \tilde{\boldsymbol{S}})||P_\theta(Y|\boldsymbol{X} \odot \tilde{\boldsymbol{S}})) = \mathbb{E}_{X,Y,\tilde{S}} \log \frac{P_{\theta*}(Y|\boldsymbol{X} \odot \tilde{\boldsymbol{S}})}{P_\theta(Y|\boldsymbol{X} \odot \tilde{\boldsymbol{S}})}$$

we can conclude that

$$\mathbb{E}_{X,Y,\tilde{S}} - \log P_{\theta^*}(Y|\boldsymbol{X} \odot \tilde{\boldsymbol{S}}) \leq \mathbb{E}_{X,Y,\tilde{S}} - \log P_\theta(Y|\boldsymbol{X} \odot \tilde{\boldsymbol{S}}).$$

If we consider the negative log likelihood of the target given the observations (i.e. $-\log P_\theta(Y|\boldsymbol{X} \odot \tilde{\boldsymbol{S}})$) as a loss function $L$ (which encodes the classification or regression function $f_{\boldsymbol{\theta}}$), then we see that minimizing the risk approximately maximizes the MI objective in Eq. 6.

## E    DETAILS OF REGULARIZATION TERM

Here we provide the detail description of the regularization term. For the vector of stochastic gates $z \in \mathbb{R}^D$, the regularization term is expressed as follows:

$$
\begin{aligned}
\mathbb{E}_Z \|\mathbf{Z}\|_0 &= \sum_{d=1}^{D} \mathbb{P}[z_d > 0] = \sum_{d=1}^{D} \mathbb{P}[\mu_d + \sigma_d \epsilon_d > -\frac{1}{2}] \\
&= \sum_{d=1}^{D} \{1 - \mathbb{P}[\mu_d + \sigma_d \epsilon_d \le -\frac{1}{2}]\} \\
&= \sum_{d=1}^{D} \{1 - \Phi(\frac{-\frac{1}{2} - \mu_d}{\sigma_d})\} \\
&= \sum_{d=1}^{D} \Phi(\frac{\mu_d + \frac{1}{2}}{\sigma_d}).
\end{aligned}
$$

The derivative of the regularization term with respect to the distribution parameter $\mu_d$ is simply the Gaussian PDF:

$$
\frac{\partial}{\partial \mu_d} \mathbb{E}_Z \|\mathbf{Z}\|_0 = \frac{\partial}{\partial \mu_d} \Phi(\frac{\mu_d + \frac{1}{2}}{\sigma_d}) = \frac{1}{\sqrt{2\pi\sigma_d^2}} e^{-\frac{(\mu_d + \frac{1}{2})^2}{2\sigma_d^2}}.
$$

The effect of $\sigma$ can be understood by looking at the value of $\frac{\partial}{\partial \mu_d}\mathbb{E}_Z\|\mathbf{Z}\|_0$. In the first iteration during the training, $\mu_d$ is 0. Therefore, during the initial phase of training, it is close to $\frac{\lambda}{\sqrt{2\pi\sigma_d^2}} e^{-\frac{1}{8\sigma_d^2}}$. In order to remove irrelevant features, this term has to be greater than the derivative of the loss with respect to $\mu_d$ because otherwise $\mu_d$ is updated in the incorrect direction. To encourage such behavior, we set $\sigma = 0.5$, which is around the maximum of the gradient during the initial phase as shown in Fig. 4. Although the point that attains the maximum moves as $\mu$ changes, we empirically observe that setting $\sigma = 0.5$ performs well in our experiments when the regularization parameter $\lambda$ is appropriately tuned (see Section I).

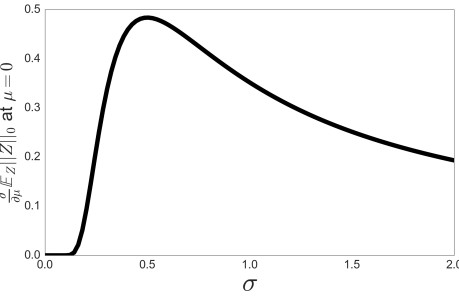

Figure 4: The plot of $\frac{\partial}{\partial \mu}\mathbb{E}_Z\|\mathbf{Z}\|_0|_{\mu=0} = \frac{1}{\sqrt{2\pi\sigma^2}} e^{-\frac{1}{8\sigma^2}}$ for $\sigma = [0.001, 2]$.

## F    ISSUES IN GRADIENT ESTIMATION OF DISCRETE RANDOM VARIABLES

In Section 3.1, we have introduced Bernoulli random $\tilde{s}_d, d = 1, ..., D$ variables with corresponding parameters $\pi_d$ into the risk objective (Eq. 4). Taking the expectation over the $\ell_0$ norm of $\tilde{\mathbf{S}}$ boils down to the sum of the Bernoulli parameters $\pi_d$. However, the optimization of the resulting objective suffers from high variance due to the discrete nature of $\tilde{\mathbf{S}}$. Here, we attempt to convey this problem by analyzing the risk term in the objective in Eq. 4. Using the Bernoulli paramterization the empirical risk $\hat{R}(\boldsymbol{\theta}, \boldsymbol{\pi})$ is expressed as

$$\sum_{\boldsymbol{z}:\{0,1\}^D} \left[ \sum_{n=1}^{N} [L(f_\theta(\boldsymbol{z} \odot \boldsymbol{x}_n), \boldsymbol{y}_n] \prod_{d=1}^{D} \pi_d^{z_d}(1-\pi_d)^{1-z_d} \right].$$

In practice, as the outer sum involves enumerating $2^D$ possibilities of the indicator variables, one can replace the outer sum with Monte Carlo samples from the product of Bernoulli distributions $B(\boldsymbol{z}|\boldsymbol{\pi})$. However, a Monte Carlo estimate of $\frac{\partial}{\partial \pi_d}\hat{R}(\boldsymbol{\theta}, \boldsymbol{\pi})$ suffers from high variance. To see this, consider the following exact gradient of the empirical risk with respect to $\pi_d$, which is

$$\sum_{\boldsymbol{z}:\{0,1\}^D, z_d=1} \left[ L(\boldsymbol{z})p_{z_{i \neq d}} \right] - \sum_{\boldsymbol{z}:\{0,1\}^D, z_d=0} \left[ L(\boldsymbol{z})p_{z_{i \neq d}} \right],$$

where $p(z_{i \neq d}) = \prod_{i \neq d}^{D} \pi_i^{z_i}(1-\pi_i)^{1-z_i}$, by absorbing the model $f_\theta(\cdot)$ and the data into $L(\cdot)$. Due to the discrete nature of $\boldsymbol{z}$, we see that even the sign of the gradient estimate becomes inaccurate if we can only access a small number of Monte Carlo samples. While a score-function estimator such as REINFORCE (Williams, 1992) can be used, it is known that the reparametrization trick reduces the variance more in practice.

## G    COMPARISON TO HARD-CONCRETE DISTRIBUTION

To evaluate the strength of the proposed continuous relaxation of the Bernoulli distribution, described in Subsection 3.1, we compare it with the Hard-Concrete distribution (Louizos et al., 2017), another continuous surrogate for Bernoulli distributions, which was originally developed for neural network model compression. The details of the Hard-Concrete distribution is described below.

### G.1    HARD-CONCRETE DISTRIBUTION

The authors in (Louizos et al., 2017) introduce a modification of Binary Concrete, whose sampling procedure is as follows:

$$u \sim U(0,1), L = \log(U) - \log(1-U)$$
$$s = \frac{1}{1 + \exp(\frac{-(\log \alpha + L)}{\beta})}$$
$$\bar{s} = s(\zeta - \tau) + \tau$$
$$z = \min(1, \max(0, \bar{s}))$$

where $(\tau, \zeta)$ is an interval, with $\tau < 0$ and $\zeta > 1$. This induces a new distribution, whose support is $[0,1]$ instead of $(0,1)$. With $0 < \beta < 1$, the probability density concentrates its mass near the end points, since values larger than $\frac{1-\tau}{\zeta-\tau}$ are rounded to one, whereas values smaller than $\frac{-\tau}{\zeta-\tau}$ are rounded to zero.

The CDF of $s$ is

$$Q_s(s|\beta, \log \alpha) = \text{Sigmoid}((\log s - \log(1-s))\beta - \log \alpha) \tag{9}$$

and so the CDF of $\bar{s}$ is

$$Q_{\bar{s}}(\bar{s}|\phi) = \text{Sigmoid}((\log(\frac{\bar{s}-\tau}{\zeta-\tau}) - \log(1 - \frac{\bar{s}-\tau}{\zeta-\tau}))\beta - \log \alpha \tag{10}$$

where $\phi = (\beta, \log \alpha, \zeta, \tau)$

Now, the probability of being the gate $z_i$ being active is $1 - Q_{\bar{s}}(0|\phi)$ and can be written as

$$1 - Q_{\bar{s}}(0|\phi) = \text{Sigmoid}(\log \alpha - \beta \log \frac{-\tau}{\zeta}) \tag{11}$$

## G.2 THE EFFECT OF THE HEAVY TAIL DISTRIBUTION

The main difference between our proposed distribution (STG) and the Hard-Concrete (Louizos et al., 2017) distribution is that the latter is based on the logistic distribution, which has a heavier tail than the Gaussian distribution we have employed. As shown in Fig 5, the heavy-tailness results in instability during training. Furthermore, our method converges much faster and more reliably than the feature selection method using the Hard-Concrete distribution on a the two-moons, XOR and MADELON datasets (see Subsection J.1 and J.2). A similar phenomenon is also demonstrated in the MNIST experiment (see Subsection J.3).

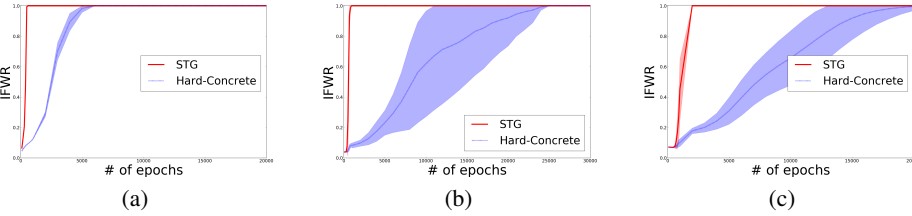

Figure 5: Comparison between STG and Hard-Concrete on the Two-Moon dataset (a), XOR (b) and MADELON (c). The shaded area represents the standard deviation, calculated by running the same experiment 10 times with different random initializations.

## H  NON-CONVEX DETERMINISTIC REGULARIZATION

Our proposed objective for feature selection (summarized in Eq. 5) consists of three major differences compared to the LASSO formulation.

- Non-linearity, which is obtained by learning parameters of a multi-layer network with non linear activation's.
- Feature gates, these are a specific form on non convex regularization.
- Stochasticity, which is achieved by injecting noise via the reparametrization trick.

In section 6.1, we have demonstrated that even in the linear regime the proposed formulation outperforms the LASSO in the task of identifying active features (see Fig. 1(a)).

The stochastic property of the gates proposed in this paper is justified using a MI prospective (see Section 4). Nonetheless, in this section we evaluate whether the practical performance gain of our method stems from the non-convex regularization or from it's combination with the injected noise.

We define a deterministic gate using $\tilde{z}_d = g(\mu_d) = \max(0, \min(1, \mu_d + 0.5))$, where $\mu_d$ is a parameter learned for each feature $d = 1, ..., D$. In the linear regression setting we can define the non-convex deterministic (NCD) as

$$\min_{\boldsymbol{\theta}, \boldsymbol{\mu}} \frac{1}{N} \sum_{n=1}^{N} (\boldsymbol{\theta}^T \boldsymbol{x}_n \odot \tilde{\boldsymbol{z}} - y_n)^2 + \lambda \sum_{d=1}^{D} \Phi \left( \frac{\mu_d + 0.5}{0.5} \right), \tag{12}$$

where $\boldsymbol{\Phi}$ is the standard Gaussian CDF. Combined with $\tilde{z}_d$ this non-convex regularized objective is deterministic and differentiable, and its solution can be approximated via gradient decent. We evaluate the applicability of such alternative objective for the task of feature selection in linear regression. The experimental setting follows the description in Section 6.1 using $D = 64$.

In Fig. 6(b), we compare the values of the deteministic gates (DNC) and stochastic gates (STG) using $N = 60$. As depicted from this figure, the STG correctly identifies the active features (first 10) and sparsifies the non-active features. The deterministic version DNC does induce partial sparsity, but fails to remove some of the irrelevant features. Furthemore, it does not push the gates values of active features up to 1. Clearly, in the deterministic setting, shrinking the value of $\mu_d$ while compensating with a large coefficient in $\boldsymbol{\theta}$ is preferable for the objective in Eq. 12.

To evaluate the probability of support recovery using DNC, we apply additional thresholding to the gates. Specifically, we use a threshold of $0.5$ to define if a feature is active or not. A comparison with STG, HC and LASSO in this setting appears in Fig. 6(a). The DNC is superior to LASSO, but inferior to the HC and STG. Which means that there is a true advantage of stochacity in our formulation.

One aspect that is crucially difference between the STG and DNC is what we call "second chance". For the deterministic formulation, if a feature is zeroed out in an early training phase, the gradient of the feature vanishes for further training and it will *never* be pulled back. For the STG, even if the feature has been pulled to zero, the gate may still be pushed back up to one at a later training phase. This is due to the injected noise which allows to *reevaluate* the gradient of each gate. In Fig. 6(c), we demonstrate this "second chance" effect using $N = 60$ samples and presenting the gate's values (throughout training) for an active feature.

To conclude, this form of deterministic gates (Eq. 12) is not suitable for an embedded feature selection method. Using thresholding seems somewhat effective; however such thresholding was not required for the STG in any of our experiments. While the shrinkage phenomenon might be avoided using further regularization on $\boldsymbol{\theta}$ this is out of the scope of the current study.

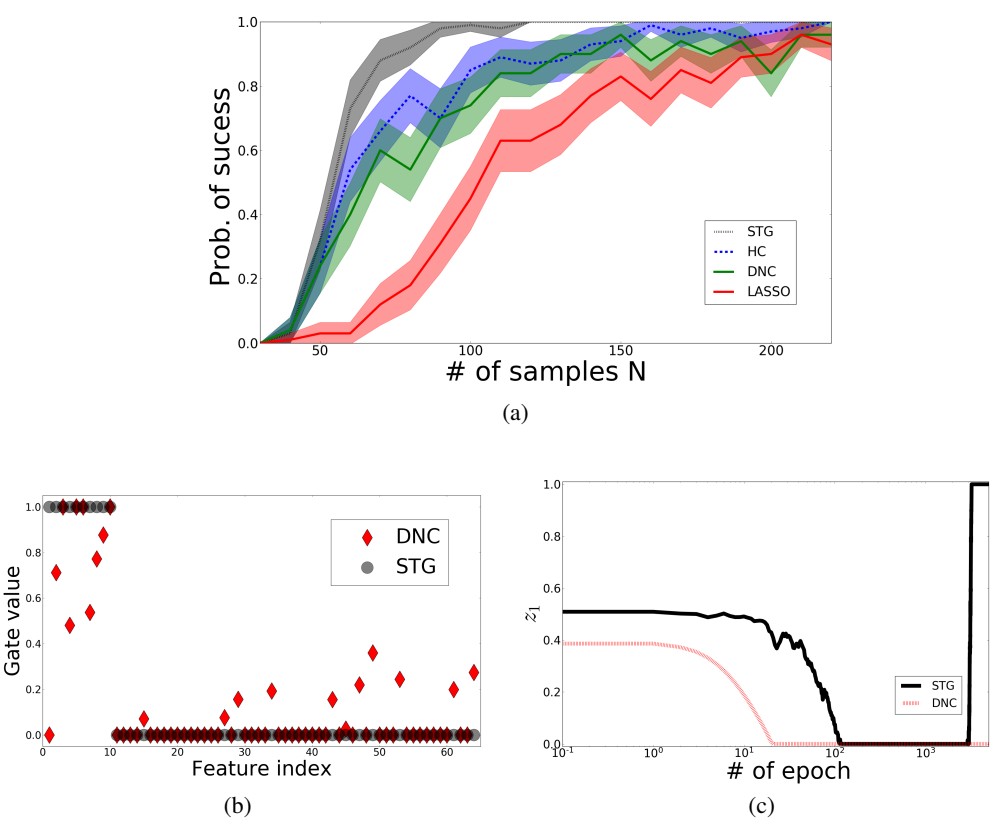

Figure 6: Support recovery in the linear regression (see Section 6.1), the dimension $D = 64$. (a) Phase transition comparison between the STG, HC, LASSO and the deterministic non convex (DNC). (b) An example of the output gates, STG ($z_d$) and DNC ($\tilde{z}_d$) for $N = 60$. Each marker represents the gate value of the corresponding feature, of which the first 10 features are active. The deterministic gates are not pushed towards $1$. (c) Demonstration of the "second chance" inherit to the STG. In the deterministic setting if a feature is eliminated its gradient vanishes for the rest of the training. The STG reevaluates this features and "recovers" after ~4000 epochs.

# I  TUNING THE REGULARIZATION PARAMETER $\lambda$

There have been several studies (Lederer & Müller, 2015; Coen et al., 2006) on the problem of tuning regularization parameter in regression and classification. Perhaps the most related regularized objective is the LASSO formulation. Studies such as (Chand, 2012; Fan & Tang, 2013) develop methods for tuning the regularization parameter in LASSO. In Eq. 5 as in the LASSO formulation, increasing the value of $\lambda$ effectively forces a sparser solution with less features. For simplicity, we focus on one popular way for tuning $\lambda$ in the LASSO. The idea is to minimize the generalization error using a cross validation procedure over a range of $\lambda$ values.

In this section, we demonstrate that this procedure is also applicable for the proposed approach. We apply the STG to the RAI dataset and evaluate the MSE using a 5-folds cross validation procedure. We compute the MSE on the validation data using 100 values of $\lambda$ logarithmically scaled in $[10^{-3}, 10^{1}]$. As demonstrated in Fig. 7, using a cross validation procedure for tuning $\lambda$ provides a clear optimal value for this example.

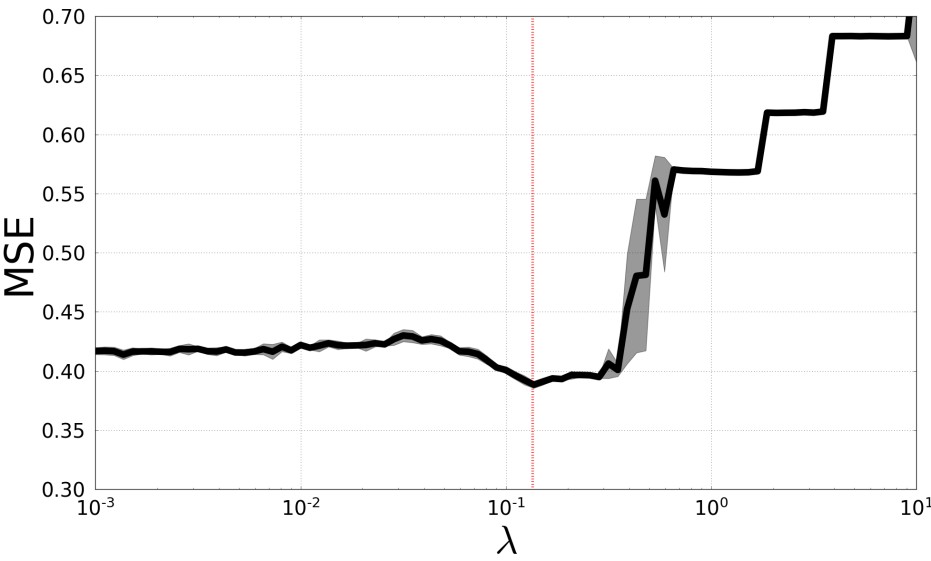

Figure 7: Demonstrating a cross validation procedure on the RAI dataset (see section I for details). We perform $5$-folds cross validation and evaluate the MSE on the test test. The optimal $\lambda = 0.135$ which seems stable in this example based on the low standard deviation.

# J  ADDITIONAL EXPERIMENTS

## J.1  MADELON DATASET

The MADELON dataset, first suggested for the NIPS 2003 feature selection problem, is a multivariate highly nonlinear binary classification problem. The MADELON dataset is generated using 32 groups of data points placed on a $5$ dimensional hyper-cube and randomly labeling them by one of the two class labels. The first $5$ informative features are then used to create $15$ additional coordinates that are formed based on a random linear transformation of the first $5$. A Gaussian noise $N(0,1)$ is added to each feature. Next, additional $480$ nuisance coordinates are added in the same manner. These features have no effect on the class label. Finally, $1\%$ of the labels are flipped. To be consistent with the XOR and two moons experiments, we use $1,500$ points from this dataset, and evaluate our proposed method in terms of its predictive power and ability to detect the informative features. We vary the regularizaton parameter $\lambda$ in the range $[0.01, 10]$ and evaluate the classification accuracy using 5 folds cross validation. In this example, we restrict our comparison to Random Forest and LASSO. We focus on Random Forest as it was the strongest competitor to our method in all of our experiments, while LASSO is evaluated because it is a widely used embedded feature selection method.

As evident from Fig. 8(a), our method achieves the highest accuracy while using less features. Moreover, as depicted from this figure, peak performance occurs when selecting 5 features, thus, our method provides a clear indication to the true number of informative features. Both LASSO and RF on the other hand, do not provide a clear indication of the true number of relevant features. In Fig. 8(b), we evaluate the effect of $\lambda$ on the number and quality of selected features. As shown in this plot, there is a wide range of $\lambda$'s such that our method only selects relevant features. Finally, as evident from the plato on the right hand side of the red plot, there is a range of $\lambda$'s such that exactly 5 features are selected.

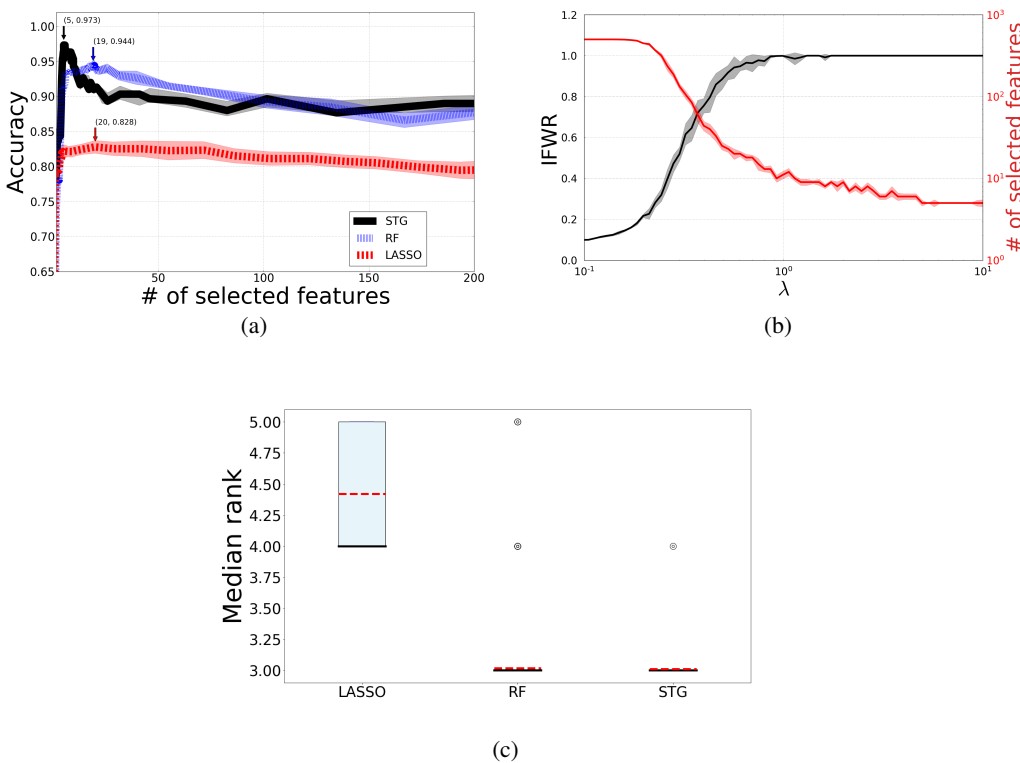

Figure 8: (a) Classification accuracy on the MADELON data sets. We evaluate performance using 5-fold cross validation for different number of selected features. In this dataset, only the first 20 coordinates are informative. In that regime the proposed method outperforms RF and LASSO. (b) An empirical evaluation of the effect the regularization parameter $\lambda$. The IFWR and the number of selected features are presented on both sides of the $y$-axis of this plot. For both plots, the mean is presented as a solid/dashed line, while the standard deviation is marked as a shaded color around the mean. (c) Box plots for the median rank of the 5 original informative features. Black line and dashed red represent the median and mean of each method. Optimal median rank in this experiment is 3.

## J.2    TWO MOONS CLASSIFICATION WITH NUISANCE FEATURES

In this experiment, we construct a dataset based on "two moons" shape classes, concatenated with noisy features. The first two coordinates $x_1, x_2$ are generated by adding a Gaussian noise with zero mean and the variance of $\sigma_r^2 = 0.1$ onto two nested half circles, as presented in Fig. 9(a). Nuisance features $x_i, i = 3, ..., D$, are drawn from a Gaussian distribution with zero mean and variance of $\sigma_n^2 = 1$. We reserve the $70\%$ as a test set, and use $10\%$ of the remaining training set as a validation set. We follow the same hyperparameter tuning procedure as in the XOR experiment. The classification accuracy is in Fig. 9(b). Based on the classification accuracies, it is evident that for a small number of nuisance dimensions all methods correctly identify the most relevant features. The proposed method (STG) and Random Forest (RF) are the only methods that achieve near perfect classification accuracy

for a wide range of nuisance dimensions. The other NN based method (DFS) seem to converge to sub-optimal solutions. We note that the median rank for all the methods is 1.5.

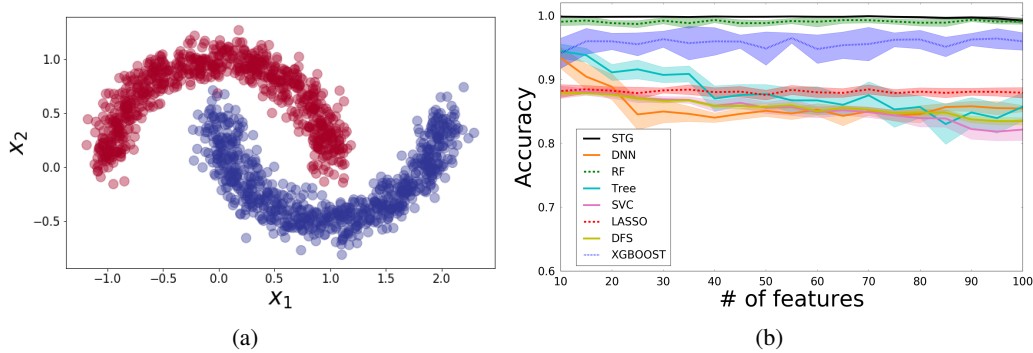

(a)                                                    (b)

Figure 9: (a) Realizations from the "Two moons" shaped binary classification class. $X_1$ and $X_2$ are the relevant features, $X_i, i = 3, ..., D$ are noisy features drawn from a Gaussian with zero mean and variance of $1$. (b) Classification accuracy (mean and standard deviation based on 20 runs) vs. the number of irrelevant noisy dimension.

## J.3   Sparse Handwritten digits classification

In the following toy example, we attempt to distinguish between images of handwritten digits of 3's and 8's using samples from MNIST (LeCun et al., 1998). The orientation and location of the digits is more or less the same throughout this dataset, therefore for these two classes (3's and 8's), we expect that some of the left side features (pixels) would be sufficient for the separation. The experiment is performed as followed. We reserve $90\%$ of the data as the test set, and train on the remaining $10\%$. We then apply STG and evaluate the classification accuracy and the number of selected features. We use the architecture [200, 50, 10] with tanh activations. The experiment was repeated 10 times, the extracted features and accuracies were consistent over 20 trials. We noticed a relatively small number of selected features, which are positioned southwest and close to the center of the images, achieve very high classification accuracy. An example of 9 randomly selected samples overlaid with the weights of the selected features is presented in Fig. 10(a). Furthermore, we also evaluate the effect of $\lambda$ on the the number of selected features and accuracy of the method. We apply our method (STG) and its variant using the Hard-Concrete distribution to a randomly sampled training set of size $N = 1500$ and vary $\lambda$ in the range of $[0.001, 0.01]$. In Fig. 10(b) we present the accuracy and sparsity level vs. the $\lambda$ parameter. This experiment demonstrates the improved performance of the proposed distribution compared to the Hard-Concrete (HC (Louizos et al., 2017)), which was designed for neural net model compression. Not only that the overall accuracy is superior, but also it seems that the transition as a function of $\lambda$ is smoother, which suggests that the method is less sensitive to the choice of $\lambda$.

## J.4   Gisette dataset

The Gisette is a handwritten dataset also appears in the NIPS 2003 feature selection challenge. The data consists of $13,500$ handwritten digits of '4' and '9'. The original digits have $28 \times 28$ pixels, which were modified for the feature selection challenge. Specifically, a random subset of the features are embedded in a 2500 dimensional space and additional 2500 irrelevant probes are added. The train, validation and test dimensions are $6000/1000/7500$ respectively. We apply the proposed STG based feature selection method and compare its performance to Random Forests (RF) and Extremely Randomized Trees (Tree). As evident from Fig. 11, we obtain a high accuracy even for a dramatic reduction in the feature size. LASSO is not presented in this experiment as its performance is dramatically inferior to the alternatives (accuracy $< 0.6$).

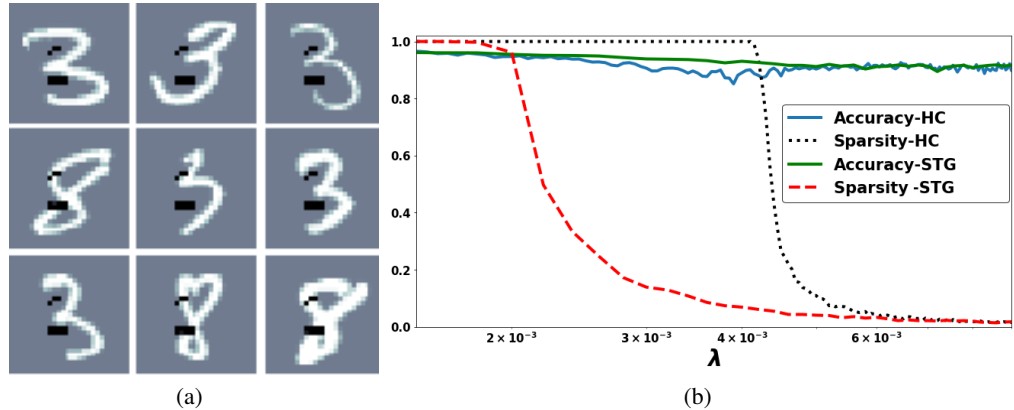

|     |     |
| --- | --- |
| (a) | (b) |

Figure 10: (a) Nine samples from MNIST (white) overlaid with the subset of 13 features (black) selected by STG. Based on only these features, the binary classification accuracy reaches $92.2\%$. For these nine randomly selected samples, all the 8's have values within the support of the selected features, whereas for the 3's there is no intersection. (b) The comparison of accuracy and sparsity level performance for $\lambda$ in the range of $[10^{-3}, 10^{-2}]$ between using our proposed method (STG) and its variant using the Hard-Concrete (HC) distribution.

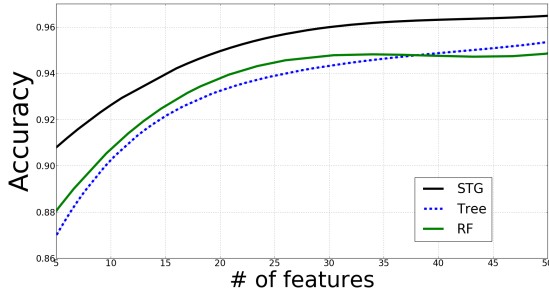

Figure 11: Classification of the binary noisy digits from the Gisette dataset. The total number of feature is 5000 of which 2500 are irrelevant probes. Here we compare the performance of the proposed approach to RF and Tree based classifier. The lines represent a least squares polynomial fit plot of the accuracy vs. number of selected features.

## J.5 REUTERS CORPUS VOLUME I

The Reuters Corpus Volume I (RCV1) consists of $800,000$ newswire stories manually labeled by $103$ categories. This is a multilable regime, i.e. each story is assigned to multiple categories. Here we focus on a binary subset of this corpus, with $23,203$ stories. The total number of feature is $47,236$ and the train, validation and test portions are $10\%, 8.5\%$, and $81.5\%$ respectively. We evaluate the performance of our method using a 5-fold cross validation. A comparison the LASSO and RF appears in Fig. 12. This example demonstrates that our method is also effective in an extremely high dimensional regime of non linear function estimation.

## J.6 DETAILS OF COX PROPORTIONAL HAZARD MODEL

Survival times are assumed to follow a distribution, which is characterized by the survival function $S(t) = P(T > t)$. A hazard function, which measures the instantaneous rate of death, is defined by $h(t) = \lim_{\Delta t \to 0} \frac{P(t < T \le t + \Delta t | T > t)}{\Delta t} = \frac{p(t)}{S(t)}$. We can relate the two functions in the following way: $S(t) = e^{-\int_0^t h(t)dt}$.

Proportional hazard models assume a multiplicative effect of the covariates $x$ on the hazard function such that

$$h(t|\boldsymbol{x}) = h_0(t)e^{\boldsymbol{\theta}^T \boldsymbol{x}},$$

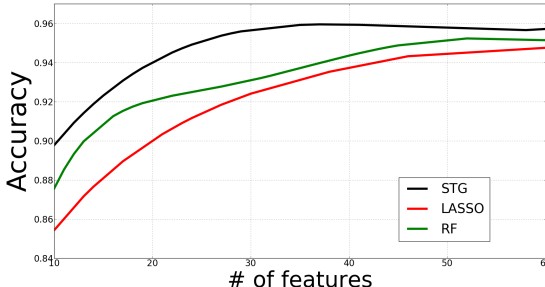

Figure 12: Classification of the version of the RCV1 textual dataset. The total number of feature is $47,236$. Here we compare the performance of the proposed approach to RF and LASSO. The lines represent a least squares polynomial fit plot of the accuracy vs. number of selected features.

where $h_0(t)$ is a baseline hazard function, which is often the exponential or Weibull distribution, and $\boldsymbol{\theta}$ is the parameter of interests.

One of the difficulties in estimating $\boldsymbol{\theta}$ in survival analysis is that a large portion of the available data is censored. However, in order to obtain estimates, Cox observed that it is sufficient to maximize the partial-likelihood, which is defined as follows: $L(\boldsymbol{\theta}) = \prod_{T_i \text{ uncensored}} \frac{e^{\boldsymbol{\theta}^T \boldsymbol{x}_i}}{\sum_{T_j \geq T_i} e^{\boldsymbol{\theta}^T \boldsymbol{x}_j}}$.

In (Katzman et al., 2018), the authors propose DeepSurv, which uses a deep neural network model to replace the linear relations between the covariate $\boldsymbol{x}$ and $\boldsymbol{\theta}$, demonstrating improvements of survival time prediction over existing models such as CPH and the random survival forest (Ishwaran & Kogalur, 2007), (Ishwaran et al., 2008).

The Molecular Taxonomy of Breast Cancer International Consortium (METABRIC) dataset consists of gene expression data and clinical features for $1,980$ patients, and $57.72\%$ have an observed death due to breast cancer with a median survival time of 116 months.

The METABRIC dataset involves 24,368 features (genes). Most genes are irrelevant for outcome prediction. To demonstrate the advantage of STG in the context of survival analysis, we selected 16 well-known genes relevant for survival (out of the 24,368 genes) that correspond to the Oncotype DX test, a gene panel used for treatment decision making. We also include five additional clinical features (hormone treatment indicator, radiotherapy indicator, chemotherapy indicator, ER-positive indicator, and age at diagnosis). We then added 200 additional irrelevant gene variables that we selected randomly from the remaining list of genes.

After we omit the null values, the number of samples is 1969. We reserve the $20\%$ for test, and use the $20\%$ of the remaining training set as validation. (That is, Train: 1260, Valid: 315, Test: 394 samples.)

**Experimental Detail**  For DeepSurv, we manually select the architecture using the validation set so that we obtain a similar performance reported in (Katzman et al., 2018). The learning rate decay is set to 1. The learning rate and the regularization parameter are optimized via Optuna using the validation set, where the search range is set $LR : [1e - 3, 1]$ and $\lambda : [1e - 3, 1]$. The hyperparameters used in the experiment are the following: architecture :[60, 20, 3], activation: selu (as suggested by (Katzman et al., 2018)), learning rate : 0.152, learning rate decay: 1.0, $\sigma : 0.5$, $\lambda : 0.023$, training epoch: 2000. Note that to see the effect of feature selection, we used the hyperparameters optimized for DeepSurv to test our method (STG-DeepSurv).

### J.7  ADDITIONAL EXPERIMENTAL DETAILS

Here we provide a full description of the procedures we have performed in the experimental parts of the paper.

For synthetic datasets are first split into train, validation and test. Validation is always $10\%$ of the train, while the exact ratios between train and test is detailed for each experiment separately. All the neural network weights are initialized by drawing from $\mathcal{N}(0, 0.1)$ and bias terms are set to 0. All the batch sizes are equal to the number of training samples. In table 3, we detail the search range

of hyperparamters as well as the exact values used in our experiments. We set n-trials = 1000 for Optuna, which is a define-by-run based hyperparameter optimization software equipped with efficient search and pruning strategies. We use SGD for all the experiments, except for the Cox model where we use Adam. All the experiments are conducted using Intel(R) Xeon(R) CPU E5-2620 v3 @2.4Ghz x2 (12 cores total).

Table 3: List of the search range for the hyperparameters used in our expirements for XOR and Two-Moon

| Param | Search range |
|---|---|
| # dense layers | [1,3] |
| # hidden units | [10, 500] |
| activation | [tanh, relu, sigmoid] |
| LR | [1e-4, 1e-1] |
| n-epoch (DFS, SG-L1-NN) | [50, 20000] |
| $\alpha$ (SG-L1-NN) | [1e-3, 1] |
| $\lambda$ (SG-L1-NN) | [1e-7, 1] |
| $\lambda$ (STG, DFS) | [1e-3, 1] |
| $\lambda$ (LASSO) | [0.01, 1] |
| n-est (RF, XGBoost, Tree) | [5,100] |
| n-boost-round (XGBoost) | [1,100] |
| Thresh (RF, XGBoost, Tree) | [0.01,0.5] |
| max-depth (XGBoost) | [0.01,0.5] |
| c (SVC) | [1e-7, 1] |

For the Phase Transition experiment, we use 0.1 as a learning rate. For all the experiments when we use Tree and RF, we use the default value for max-depth, so that nodes are expanded based on the purity. For the XOR problem, the exact architectures used for the NN based methods are: (STG/HC): $[476, 490, 14]$ with Tanh, (DFS): $[100, 10]$ with Tanh, (SG-L1-NN): $[100, 10, 5]$ with Tanh. For the two moons we use (STG): $[490, 406, 18]$ with Tanh, (DFS): $[158, 27, 224]$ with Tanh, (SG-L1-NN): $[88, 28, 27]$ with Tanh. For the XOR problem, we attempted to use Optuna to optimize parameters of DFS and SG-L1-NN, but we ended up using the architecture suggested by the authors (Li et al., 2016; Scardapane et al., 2017) as they outperform the values suggested by Optuna. The number of epochs used for the XOR problem is $20K, 14K, 800$ for STG/HC, DFS and SG-L1-NN respectively. Regularization parameters are $0.17, 3.3e - 5$ and $3e - 5$ respectively. The number of epochs used for the two-moons problem is $20K, 1570, 708$ for STG/HC, DFS and SG-L1-NN respectively. Regularization parameters are $0.48, 9e - 3$ and $1e - 3$ respectively. We note that the regularization parameters and learning procedure is different in nature, as we use an $\ell_0$ type penalty. For the PBMC experiment, the architecture was hand-tuned using the validation set and set as $[200, 100, 50, 10]$ with Tanh. Learning rate was 0.2 and the number of epochs 4000. The hyperparameter $\lambda$ varies in the range $[0.001, 0.11]$ to achieve different levels of sparsity. For MADELON, we use the architecture optimized for the binary XOR classification. The number of epochs used is $20K$, the learning rate is 0.06 and $\lambda$ varies in the range $[0.01, 10]$. For regression, the learning rate and the $\lambda$ are optimized via Optuna using the search range $LR : [1e - 4, 1]$ and $\lambda : [1e - 4, 10]$ based on validation. The parameters used are the following: (SE1) architecture [600, 200, 100, 50, 1] with ReLu activations, num epochs: 5, $\lambda : 5$, $LR : 0.0001$ (SE2) architecture [600, 300, 150, 60, 20] with ReLu activations, num epochs=2000, $\lambda : 5$, $LR : 0.001$ (SE3) architecture [600, 300, 150, 60, 20] with ReLu activations, num epochs : 1000, $\lambda : 1$, $LR : 0.005$. (RCP) architecture [1000, 300, 150, 60, 20] with ReLu activation, num-epochs: 2000, $\lambda : 5.0$, $LR : 0.001$. (REL) architecture [26,91,63] with ReLu activation, num-epochs: 1600, $\lambda : 0.031$, $LR : 0.007$. (RAI) architecture [10,177] with ReLu activation, num-epochs: 1800, $\lambda : 0.019$, $LR : 0.002$. Architectures for SE1-SE3 and RCP where tuned manually.

The ratio of train/test/valid split is 1:1:1 for synthetic data. For the real data (RCP and REL), the train size is 6000, the test and valid size is 1000 samples. For RAI the train size is 5000, the test and valid size is 1000 samples.

In order to define the IFWR, for STG, the $d^{th}$ feature weight is set to $\max(0, \min(1, \mu_d + 0.5))$. For other neural net based methods, it is given by $\sum_j W_{dj}$, where $W$ is the weight matrix of the first

layer. For other methods we just used the feature relevance returned by the trained model. Finally, the LASSO's IFWR in the XOR experiment was omitted from the manuscript as it suffered from high variance.

Regarding the comparison performed in the two-moons and XOR problem, we believe that adding IFWR along with classification accuracy versus number of feature selected provides a complementary perspective in demonstrating the efficacy of feature selection techniques. We emphasize that our goal is not to just rank features but select features by assigning the weight of $0$ to irrelevant features while simultaneously obtaining good predictive accuracy.

