# OpenReview forum: "Feature Selection using Stochastic Gates"
_ICLR.cc/2020/Conference — Reject_

### Official Review · AnonReviewer1 · 2019-10-23
**Official Blind Review #1**

**Rating:** 6

**Review:**

The author rebuttal sufficiently addresses my concerns, so I am upgrading my score.

***

The paper considers the problem of embedded feature selection for supervised learning with nonlinear functions. A feature subset is evaluated via the loss function in a "soft" manner: a fraction of an individual feature can be "selected". Sparsity in the feature selection is enforced via a relaxation of l0 regularization. The resulting objective function is differentiable both in the feature selection and learned function making (simultaneous) gradient-based optimization possible. A variety of experiments in several supervised learning tasks demonstrates that the proposed method has superior performance to other embedded and wrapper methods.

My decision is to reject, but I'm on the fence regarding this paper. I'm not clearly seeing the motivation for an embedded feature selection method for neural network models: for the datasets considered in the paper, it would seem that training a nonlinear model that used all the features would result in performance at least as good as training the nonlinear model with a prepended STG layer. Perhaps there is evidence that filtering features, e.g., irrelevant features, results in higher accuracy and that the prepended STG layer achieves this accuracy, but that evidence is missing from the paper. Also, there could be downstream computational savings, e.g., at prediction time, if the dimension was very large, but this is not the setting tested in the experiments. I suppose interpretability could be considered motivation, but, even so, isn't there at least one simpler, deterministic approach (described below) that also "solves" the problem? Finally, it isn't clear how the method scales with increasing sample size and dimension as all the datasets tested are relatively small in these respects.

***

Questions and suggestions related to decision:

* The performance values using all features should be included in the experimental results so that the value added by STG can be assessed.

* Why not use the simpler deterministic and differentiable relaxation z = \sigma(\mu), where \sigma() is a "squashing" function from the real numbers to [0,1] applied element-by-element to the vector \mu? What specifically is/are the advantage(s) that the randomness in the definition of z at the bottom of pg. 3 provide over this deterministic alternative?

* Though well-described and methodologically rigorous, the experimental comparison is none-the-less a little disappointing: one dataset for classification and half the datasets for regression are synthetic and low-dimensional. The remaining regression datasets are real but also low-dimensional. The survival analysis dataset is also low-dimensional (as described in the supplementary material). This leaves one real classification dataset which was on the order of 20,000 examples and 2500 features. Why were larger sample-size and dimensionality datasets not tested? These should be readily available. For example, the gisette dataset from the NIPS 2003 feature selection challenge has 5000 features. See "MISSION: Ultra Large-Scale Feature Selection using Count-Sketches" by Aghazadeh & Spring et al. (2018) for other high-dimensional datasets. Even a single run for each large dataset would have provided some evidence of scalability.

***

Other minor comments not related to decision:

* "Concrete Autoencoders for Differentiable Feature Selection and Reconstruction" by Abid et al. (2019) targets unsupervised feature selection but has enough similarities in the approach that it should be considered related work.

* [Typo?] The unnumbered equation after (5) should not have a sum over d in the second term. Perhaps a sum over k was intended? Also, in this equation, the gradient of the loss wrt/ z samples, average of gradients over z samples times..., does not seem to match what the gradient would be given the algorithmic description in the supplementary material, a gradient of the (sample) average z times...

* The abstract states the paper is proposing a method for high-dimensional feature selection, but all of the experiments have datasets with max. dimensionality 2538.

* Some discussion of how the regularization parameter can be selected by a user of the proposed method would be good to include.

**Experience Assessment:**

I have read many papers in this area.

**Review Assessment: Checking Correctness Of Derivations And Theory:**

I assessed the sensibility of the derivations and theory.

**Review Assessment: Checking Correctness Of Experiments:**

I assessed the sensibility of the experiments.

**Review Assessment: Thoroughness In Paper Reading:**

I read the paper at least twice and used my best judgement in assessing the paper.

---

> ### Author Response · Authors · 2019-11-15
> **Response to Blind Review #1**
>
> We thank the reviewer for the detailed and constructive comments.
>  We propose to use a neural network for feature selection, rather than to perform feature selection in neural networks. Clearly, CNN do not require feature selection since the inputs are pixels. There still isn’t an effective method for performing feature selection while learning nonlinear complex relationships between variables. Our solution is yet the first $\ell_0$ based embedded method to achieve this task. We disagree with the reviewer's assumption that training a nonlinear model which uses all feature results in comparable performance.  In all of the examples presented in the paper, the STG improves the accuracy dramatically compared to all alternatives and compared with a similar NN without feature selection (see DNN results added to plots and tables). Regardless of generalization, identifying a small subset of features that interact through a nonlinear model leads to a number of benefits: reducing experimental costs, enhancing interpretability, computational speed up and even improving model generalization on unseen data. In biomedicine, scientists collect multitude datasets comprising of many biomarkers (e.g., genes or proteins) that require the development of effective diagnostics or prognostics models. For instance, in Genome-wide association studies (GWAS), feature selection can help identify such models and lead to improved risk assessment and reduced cost.
>
>  In what follows we address each comment following the presented order.
> P1. We have added the performance of a standard neural network to all of the relevant examples (see DNN in tables and plots). Note that in the COX example DeepSurv is, in fact, a neural network without feature selection. P2. This is a very nice suggestion. We have examined the deterministic non-convex regularization presented by the reviewer and observed that is inferior to the proposed approach in various aspects. In a new section that appears in Section H in the appendix, we detail and demonstrate the differences between the deterministic and stochastic formulation of the proposed gates.  We have observed that without stochasticity “deterministic gates” converge to values in the range of (0,1), in contrast, the  “stochastic gates” converge to {0,1}. Deterministic gates accompanied by thresholding somewhat improves this inherent problem. Importantly, stochastic gates achieve superior results to this two-step deterministic procedure as we now demonstrate. One clear additional advantage of stochastic gates is what we call a “second chance”, where the injected noise allows revaluation of features even if their parameters reached 0/1 in an early training phase.
>
> P3. Following the reviewer's concern, we have added two new experiments consisting of a large number of features (Gisette dataset (5000 features) and Reuiter Corpus Volume 1 dataset (47,236 features)). Please see section J.4 and J.5 in the Appendix in which we present the details and results of these two high dimensional experiments.
> P4. Our initial draft has been available online prior to the submission of [1]. The authors in [1] cite our original draft in their manuscript. P5. Thanks for spotting this, we have revised this unnumbered equation to correct this mistake. P6 Thanks again for this suggestion, we are now demonstrating applicability to ~50K features as pointed in in P3.  P7. We have added a new section in the appendix (see section I), discussing and demonstrating how to tune the regularization parameter.
>
>
> [1]"Concrete Autoencoders for Differentiable Feature Selection and Reconstruction" by Abid et al. [2019]

---

### Official Review · AnonReviewer2 · 2019-10-31
**Official Blind Review #2**

**Rating:** 3

**Review:**

The paper is concerned with embedding a supervised feature selection within a classification setting.
The originality is to use an L_0 regularization (counting the number of retained features), besides the classification loss; the authors leverage the ability to include boolean variables in a neural network and to optimize their value using gradient descent through the reparameterization trick.

I am mildly convinced by the paper:
* Out of the four contributions listed p. 2, STG is the most convincing one; still, the description thereof is not cristal clear: the reparametrization trick is not due to the authors. The discussion (section 5) needs be more detailed, adding the HC details (presently in appendix); could you comment upon the difference between the proposed STG and the Gumbel-Softmax due to Jang et al, cited ?
* Likewise the authors delve into details regarding the early state of the art, while omitting some key points. For instance, p. 3, the fact that many authors replaced an L_0 penalization with an L_1 one is rooted on the fact that, provided that the optimal L_0 solution is sparse enough, the L_0 and L_1 problems have same solutions. This section can be summarized;
* the sought sparsity is assumed to be known, which is bold;
* Assumption 2 is debatable; one would like to find at most the Markov blanket of the label variable. See Markov Blanket Feature Selection for Support Vector Machines, AAAI 08.
* There are digressions in the paper which make it harder to follow the argumentation (section 6.1); section 6.2 is not at the state of the art; in Guyon et al's Feature Selection Challenge (2003), the Arcene artificial problem involves a XOR with 5 key features, and 15 additional features are functions of the key features.

Suggestion, you might compare with the L_0 inspired regularization setting used for unsupervised feature selection in Agnostic Feature Selection, Doquet et al, 2019.

Details: check the citation style: use \citep instead of \cite.

**Experience Assessment:**

I have published one or two papers in this area.

**Review Assessment: Checking Correctness Of Derivations And Theory:**

I assessed the sensibility of the derivations and theory.

**Review Assessment: Checking Correctness Of Experiments:**

I assessed the sensibility of the experiments.

**Review Assessment: Thoroughness In Paper Reading:**

I read the paper at least twice and used my best judgement in assessing the paper.

---

> ### Author Response · Authors · 2019-11-15
> **Response to Blind Review #2**
>
> We thank the reviewer for the detailed and constructive comments. In the following, we address all the points raised by the reviewer.
> P1. We agree that the STG is indeed one major contribution, which is a simple yet highly effective relaxation to the Bernoulli distribution. We argue that the method itself (embedded feature selection with the STG) is of major importance. To the best of our knowledge, it provides the first embedded non-linear feature selection solution. This is analogous to the contribution of LASSO to the statistical community, despite the fact the same $\ell_1$ regularization was used earlier for basis pursuit.
> Furthermore, we believe that the STG is useful for other applications. The Hard Concrete (HC) [3] improves upon the Gumbel-Softmax (or equivalently the Concrete distribution) by applying a hard thresholding function to the Concrete values, which allowed the authors in [3] to achieve network sparsification. The proposed STG differs from the HC, as the first is based on a Gaussian and the latter relies on a uniform distribution. We have demonstrated extensively that the HC is less suitable for the task of feature selection compared to STG. This is partially due to the higher empirical variance the HC suffers from. This is demonstrated in several experiments and in Appendix G.2.  We have revised section 5 to include more details on the HC.
> P2. This point is now incorporated in section (2.2) which has been edited to be more concise.
> P3. This assumption is only used for providing a theoretical connection between the deterministic and stochastic objectives. For example, the true sparsity is used to provide a theoretical analysis of the LASSO in [1,2]. In practice, we do not need to know this number.
> P4. It is an interesting suggestion to generalize the assumption in future work.
> P5. We have rephrased subsection 6.1 to improve the flow in the experimental section. The structure of the experiments is changed, we start with experimental evaluation of the proposed approach in section 6 followed by applications in section 7. Acrene is a cancer dataset, We believe the reviewer is referring to the MADELON dataset. We have presented results using this data set in the original submission, which appears in the supplementary material. We have demonstrated that the proposed method achieves state of the art results on the MADELON data.
> P6. We would be glad to compare to this method. We have emailed the authors to request the code. P7. Thanks for pointing this out. We have changed the citation style as suggested.
>
>  [1] “Sharp thresholds for high-dimensional and noisy recovery of sparsity”, Martin J. Wainwright [2009]
> [2] "On the prediction performance of the lasso." Dalalyan, Arnak et al.  [2017]

---

### Official Review · AnonReviewer4 · 2019-11-04
**Official Blind Review #4**

**Rating:** 3

**Review:**

The authors propose a feature selection method for high-dimensional datasets that attempts to fit a model while selecting relevant features.
The strategy they follow is below:

1. They formulate feature selection as an optimization problem by augmenting standard empirical risk minimization with zero-one variables associated with each feature representing the absence-presence, and adding a penalty proportional to the number of included features. They relax the discrete variables using a continuous relaxation and provide a simple unbiased estimator for the gradient of the relaxation. After training the relaxation is rounded to a zero-one solution by a simple scheme.
2. They provide an information theoretic motivation for their formulation of feature selection
3. They exhibit the performance of their method on a number of synthetic and real data scenarios: (i) linear models with a true underlying sparse parameter, (ii) binary classification with a small number of true determining features, (iii) regression performance post-feature selection with synthetic non-linear models (with a few determining features) and two real datasets. They also use the method for a classification problem with RNA-seq data on T-cells and a survival analysis based on a breast-cancer dataset called METABRIC.

Despite my recommendation, there are a number of things that I like about the paper that I list below, along with directions where I believe the article can be improved.
1. At a certain abstraction, the main idea of the paper is to do feature selection at the same time as model fitting (as the LASSO for e.g. does) while ignoring constraints of convexity raised in the optimization, and simply using stochastic gradient with a reasonable unbiased estimate of the gradient. This is a reasonable idea, particularly if under some reasonable assumptions, the non-convex formulation that is obtained is expected to be computationally 'benign'.
2. In a number of the experiments, and particularly 6.1 (sparse linear model) 6.2 (noisy XOR classification) I suspect the non-convex formulation is what is providing a lot of the improvement. This has been observed empirically in a number of other settings, for e.g. in matrix completion/factorization problems. Verifying this hypothesis in a simple, synthetic (and therefore controlled) dataset would be a good contribution for a future version.
3. The authors have done a fairly good job of validating the method in a number of different settings, even if some of the presentation of their results can possibly be somewhat improved. For e.g. the median rank is better shown with box plots (as in the Chen et al 2018 paper cited by the authors).
4. There are a number of relaxations of discrete variables used in optimization and theoretical computer science literature. For instance, the approach of the authors is reminiscent to 'mean field' methods, or standard linear programming relaxation of combinatorial optimization problems (i.e. the first level of the Sherali-Adams LP hierarchy). On the other hand, naive versions of this are not likely to work well on (say) sparse linear regression. The current methods do which suggests that the continuous relaxation is useful.

At an expository level, I also think the paper could do with quite a bit of improvement:
1. The introduction is sparse and hurried, and without providing sufficient motivation and intuition for the contributions of the article.
2. In 6.4, 6.5, the introduction about RNA-seq or Cox models can be removed and relevant work cited instead.
3.  Organizing the experiments as real data, and synthetic data might be semantically better, though that would necessitate splitting Table 1. I am also unclear on why the authors show  performance in Tables 1, 2 independent of the number of features selected, while for the experiment on RNA-seq data the full accuracy/#features tradeoff is given. The sparse explanation about using the Optuna paper is certainly not enough.

Minor comments not related to decision:
1. The value for \alpha_N in synthetic sparse linear model experiment of 6.1 likely has an extraneous \sqrt \log k


**Experience Assessment:**

I have read many papers in this area.

**Review Assessment: Checking Correctness Of Derivations And Theory:**

I assessed the sensibility of the derivations and theory.

**Review Assessment: Checking Correctness Of Experiments:**

I assessed the sensibility of the experiments.

**Review Assessment: Thoroughness In Paper Reading:**

I read the paper at least twice and used my best judgement in assessing the paper.

---

> ### Author Response · Authors · 2019-11-15
> **Response to Blind Review #4**
>
>  We thank the reviewer for the detailed and constructive comments. The reviewer organized his recommendations in 3 groups which we address in consecutive order. P.1) We agree that one main advantage of the proposed method is its ability to perform feature selection while learning a non-linear model. This is achieved by a non-convex objective. We demonstrate empirically using several examples that this is in fact computationally ‘benign’. This is similar to other recent successful results obtained by non-convex optimization via deep neural networks. P2. Indeed, a non-convex formulation is useful for various applications. However, such formulation alone is not sufficient for an embedded feature selection method. A new section was added (see Appendix H) to evaluate the effect of such deterministic non-convex regularization. We have observed that without stochasticity, “deterministic gates” converge to values in the range of (0,1), while the “stochastic gates” converge to {0,1}. The deterministic gates accompanied by thresholding somewhat improve the problem. However, stochastic gates achieve superior results to this two-step deterministic procedure. One clear additional advantage of stochastic gates is what we call a “second chance”, where the injected noise allows re-evaluation of features even if their parameters reached 0/1 in an early training phase.  P3. Thanks for the suggestion we have added box plots of the median rank for the XOR experiment (see Fig. 1C) and MADELON (see Fig. 6C). This indeed demonstrates the superiority of our method over the alternatives. Furthermore, to evaluate our method in a high dimensional regime we experimented with two additional datasets (see Appendix J.4 and J.5). P4. We agree we believe that this type of continuous relaxation is useful for other applications as well (e.g. basis pursuit, robust representation variational inference and more). Furthermore, we demonstrate that our relaxation outperforms the previously suggested “Hard-Concrete”, not to mention the “Concrete” relaxation which fails to sparsify the feature space.
>
> We next respond to the improvement suggestions provided by the reviewer.
> S1. We have revised the introduction, providing a more concise motivation and explanation of the proposed approach.
> S2. Sections 6.4 and 6.5 (now 7.1 and 7.2) were abbreviated. We now refer the reader to the relevant citations for a description of the PBMC and COX datasets.
> S3. We have re-organized the experimental section. Section 6 now provides experiments evaluating the method in the linear and non-linear setting. In section 7, we demonstrate its utility to biomedical applications, where reducing the number of features translates to cheaper and more accurate medical assays.
> The results provided in Table 1 augment the values provided in a recent paper (SRFF) [1]. In [1],  the authors provide the optimal results (without referring to the number of features). Here, we intended to demonstrate that the proposed method competes with SRFF in their setting. We have added the number of selected features in Table 2. The description of Optuna is now expanded in the supplementary material.
> Finally, regarding the minor comment 1, we have used the value of $alpha_N$ as presented in [2] (see section IV). The expression for alpha_N in this paper includes the \sqrt \log k term.
> [1] Gregorová, Magda, et al. "Large-scale nonlinear variable selection via kernel random features." Joint European Conference on Machine Learning and Knowledge Discovery in Databases. Springer, Cham, 2018.‏
>  [2] “Sharp thresholds for high-dimensional and noisy recovery of sparsity, Martin J. Wainwright [2009]”

---

### Public Comment · ~Ian_Connick_Covert1 · 2019-09-28
**Differences with prior work**

Hi, this is nice work. However, I have a question about connections with a couple existing papers.

Can you elaborate on how your method differs from "Dropout Feature Ranking for Deep Learning Models" (Chang et al., 2017)? Your objective (Eq. 4) seems exactly the same as theirs (Eq. 2). And while Chang et al. address the problem of feature ranking, not feature selection, they also note the objective's link with l0 regularization.

The only difference I noticed is a different parameterization for the continuous relaxation of Bernoulli samples. Your stochastic gate (STG) relaxation may lead to faster convergence, but I only saw a comparison with the Hard-Concrete. I'm curious what you would expect to find in a comparison with the original Concrete relaxation.

Essentially the same method was also used in "Adaptive Compressed Sensing MRI with Unsupervised Learning" (Bahadir et al., 2019), see Eqs. 1-3.

Could you explain what differentiates this work? If nothing else, it seems those papers should be cited.

---

> ### Author Response · Authors · 2019-10-02
> **Response to comment**
>
> Thank you for bringing these two papers to our attention. We will cite these papers in the manuscript.
>
> The paper  "Dropout Feature Ranking for Deep Learning Models" uses the original concrete distribution and proposes a method for feature ranking. As opposed to the Hard Concrete distribution, the original concrete distribution does not provide sparsity. Therefore, the authors propose to rank the features and then train a new network that uses the top-ranked features. The distinction is analogous to the difference between L2 and L1 regularization for linear regression problems where the former does not sparsify the variables. Furthermore, the method proposed in their paper is a wrapper method.
>
> On the other hand, in our study, we focus on developing a fully embedded feature selection method. Specifically, we studied two candidate distributions: a) the Hard Concrete and b) our novel STG. We demonstrate that the Hard Concrete distribution results in feature sparsification, but suffers from high variance. Importantly, we empirically show that our novel STG distribution overcomes this limitation and resulting in high performance in terms of accuracy and feature selection.
>
>
>
> The paper "Adaptive Compressed Sensing MRI with Unsupervised Learning" (Bahadir et al., 2019) addresses the problem of compressed sensing of MRI scans. The authors use a trick similar to the concrete distribution in order to undersample the number of Fourier coefficients needed for reconstruction of the MRI scan. The method is unsupervised and uses a different objective and regularization than the one proposed in our study. Furthermore, our manuscript has been available online prior to the work of Bahadir et al., 2019. In fact, the most related work to the study by Bahadir, is  "Concrete Autoencoders: Differentiable Feature Selection and Reconstruction."  by Balın, Muhammed Fatih et. al 2019, which cites our original preprint.

---

### Decision · Program_Chairs · 2019-12-19

**Decision:**

Reject

**Comment:**

The authors propose a method for feature selection in non linear models by using an appropriate continuous relaxation of binary feature selection variables. The reviewers found that the paper contains several interesting methodological contributions. However, they thought that the foundations of the methodology make very strong assumptions. Moreover the experimental evaluation is lacking comparison with other methods for non linear feature selection such as that of Doquet et al and Chang et al.